# Systematic identification of 20S proteasome substrates

Monika Pepelnjak[1,3], Rivkah Rogawski[2,3], Galina Arkind[2], Yegor Leushkin[2], Irit Fainer[2], Gili Ben-Nissan [2], Paola Picotti [1✉] & Michal Sharon [2✉]

## Abstract

**For years, proteasomal degradation was predominantly attributed to the ubiquitin-26S proteasome pathway. However, it is now evident that the core 20S proteasome can independently target proteins for degradation. With approximately half of the cellular proteasomes comprising free 20S complexes, this degradation mechanism is not rare. Identifying 20S-specific substrates is challenging due to the dual-targeting of some proteins to either 20S or 26S proteasomes and the non-specificity of proteasome inhibitors. Consequently, knowledge of 20S proteasome substrates relies on limited hypothesis-driven studies. To comprehensively explore 20S proteasome substrates, we employed advanced mass spectrometry, along with biochemical and cellular analyses. This systematic approach revealed hundreds of 20S proteasome substrates, including proteins undergoing specific N- or C-terminal cleavage, possibly for regulation. Notably, these substrates were enriched in RNA- and DNA-binding proteins with intrinsically disordered regions, often found in the nucleus and stress granules. Under cellular stress, we observed reduced proteolytic activity in oxidized proteasomes, with oxidized protein substrates exhibiting higher structural disorder compared to unmodified proteins. Overall, our study illuminates the nature of 20S substrates, offering crucial insights into 20S proteasome biology.**

**Keywords** 20S Proteasome; Protein Degradation; Mass Spectrometry; Intrinsically Disordered Proteins; Proteolytic Processing
**Subject Categories** Post-translational Modifications & Proteolysis; Proteomics

## Introduction

The composition of the proteome must dynamically adapt to developmental cues, genetic changes, epigenetic marks, environmental stress and aging; challenges that all cells continuously encounter. This requirement is mediated by a network of cellular pathways that control protein synthesis, folding, trafficking, aggregation and degradation (Powers et al, 2009). The constant flux of the proteome is manifested in the fact that most proteins have a half-life several fold shorter than the cell generation time, even in rapidly dividing cells (Eden et al, 2011). Protein degradation is a key factor not only in protein turnover and controlling the levels of short-lived regulatory proteins, but also in preventing the accumulation of damaged or misfolded proteins (Rousseau and Bertolotti, 2018b). Failure or malfunction of this process may lead to various illnesses, including cancer and neurodegenerative diseases (Rousseau and Bertolotti, 2018a).

Most cellular proteins are degraded by proteasome-mediated pathways (Collins and Goldberg, 2017), which can eliminate substrates by two alternative mechanisms (Goldberg, 2003; Kumar Deshmukh et al, 2019). The first is ubiquitin- and ATP-independent, while the second is dependent on both. Ubiquitin- and ATP-independent degradation can be carried out by the free 20S proteasome complex. This particle is the actual catalytic machinery of the proteasome, wherein breakage of peptide bonds occurs (Kish-Trier and Hill, 2013). The 20S proteasome is composed of 28 subunits arranged in a cylindrical structure comprising four heptameric rings: two outer α-subunit rings (PSMA1–PSMA7) that embrace two central β-subunit rings (PSMB1–PSMB7) (Kish-Trier and Hill, 2013). Three different types of proteolytic sites are found within the 20S particle β-rings, namely chymotrypsin-, caspase- and trypsin-like activities, contained within PSMB5, PSMB6 and PSMB7, respectively. This architecture creates a compartment whose proteolytic active sites are restricted to its interior, so that only proteins entering this chamber are degraded. Thus, proteins containing unfolded or unstructured regions, which can independently enter into the 20S proteasome's narrow aperture, can be passively degraded (Ben-Nissan and Sharon, 2014a; Kumar Deshmukh et al, 2019). The 20S complex can be capped at one or both ends by the proteasome activators PA28αβ, PA28γ, and PA200 (Cascio, 2021; Pickering and Davies, 2012c). These regulators open the 20S proteasome gate and enhance the complex's catalytic activity.

Alternatively, ubiquitin and ATP-dependent degradation of folded substrates is executed by the 26S and 30S proteasomes that are formed by binding of one or two 19S regulatory complexes to the 20S proteasome, respectively (Hershko and Ciechanover, 1998). The 19S regulatory complex recognizes degradation substrates through their polyubiquitin tag and catalyzes substrate

[1]Institute of Molecular Systems Biology, Department of Biology, ETH Zurich, Zurich, Switzerland. [2]Department of Biomolecular Sciences, Weizmann Institute of Science, Rehovot 7610001, Israel. [3]These authors contributed equally: Monika Pepelnjak, Rivkah Rogawski. ✉E-mail: picotti@imsb.biol.ethz.ch; michal.sharon@weizmann.ac.il

deubiquitination, unfolding, and translocation into the 20S catalytic particle. Degradation by the 26S proteasome is coordinated by three different types of ATP-dependent enzymes (E1, E2, and E3) that ubiquitinate the substrate and sensitize it to degradation (Scheffner et al, 1995). However, degradation by the ubiquitin-dependent or -independent pathways are not mutually exclusive, and different pools of the same protein can be sent to degradation via either route.

Our understanding of the alternative, simpler degradation pathway solely mediated by the 20S proteasome significantly lags behind the extensive research on the 26S proteasome. The historical emphasis has predominantly centered on the 26S proteasome, shaping research priorities and methodologies in this area. Furthermore, commonly employed proteasome inhibitors, like MG132, are designed to affect both the 20S and 26S proteasomes (Lee and Goldberg, 1998), posing challenges in specifically studying substrates degraded exclusively by the 20S proteasome. Nevertheless, current knowledge highlights that the free 20S proteasome, unbound from its activators, constitutes the most abundant fraction of total proteasome content in various cell types (Fabre et al, 2013; Fabre et al, 2014). In addition, under oxidative conditions, cellular degradation predominantly occurs via the 20S proteasome (Grune et al, 2011; Livnat-Levanon et al, 2014; Obin et al, 1998; Reinheckel et al, 1998; Wang et al, 2010). Recent studies also indicate that the 20S proteasome degradation route is tightly regulated (Deshmukh et al, 2023; Moscovitz et al, 2015; Olshina et al, 2020) and that it influences various cellular processes, such as neuronal stimulation (Ramachandran et al, 2018; Ramachandran and Margolis, 2017), antigenic peptide production (Dalet et al, 2010; Vigneron et al, 2019) and parasite growth (Dekel et al, 2021; Sharon and Regev-Rudzki, 2021). It was found that 20S and 26S proteasomes generate different peptide products from the same target protein, with a greater variety and comparatively longer peptides produced by the 20S in comparison to the 26S proteasome (Sahu et al, 2021). These longer peptides may retain certain secondary structures, increasing their potential to act as signaling molecules (de Araujo et al, 2019). Similarly, it has been shown that the activity of 20S proteasomes is not restricted to the complete degradation of its substrate, but rather there are proteins that are cleaved by the 20S proteasome at specific sites to generate functional cleavage products (Baugh and Pilipenko, 2004; Moorthy et al, 2006; Olshina et al, 2018; Solomon et al, 2017; Sorokin et al, 2005). Taken together, these findings suggest that many cellular processes are coordinated by this ubiquitin- and ATP-independent degradation mechanism.

Current knowledge regarding the repertoire of 20S proteasome substrates mainly originates from individual case studies, which have reported the identity of specific proteins susceptible to 20S-mediated degradation (Ben-Nissan and Sharon, 2014a). For example, various signaling and regulatory proteins that contain intrinsically disordered regions (IDRs) have been shown to be degraded by the 20S proteasome, including the tumor suppressors p53, p73, and retinoblastoma protein, the proto-oncoprotein c-Fos, the cell cycle regulators p27 and p21 and the neurodegenerative disease-related proteins tau and α-synuclein (Ben-Nissan and Sharon, 2014b; Hwang et al, 2011; Pickering and Davies, 2012a). A recent effort to systematically uncover the scope of 20S proteasome substrates, adopted a distinctive approach by pre-conditioning HeLa nuclear-rich cellular extracts at 95 °C before subjecting them to degradation by purified 20S proteasomes (Myers et al, 2018). This study identified numerous 20S proteasome substrates, many of which are proteins involved in the formation of phase-separated granules. While the study exhibited a preference for heat-soluble nuclear proteins featuring IDRs, its outcomes emphasize the proposition that a substantial portion of the proteome is susceptible to 20S proteolysis. This underscores not only the critical significance of understanding the precise composition of the 20S substrate landscape but also highlights the importance of developing methodologies capable of achieving this insight within native conditions.

To systematically identify 20S proteasome substrates under native conditions, we used an advanced mass spectrometry (MS) approach coupled to biochemical and cellular analysis. Specifically, we have adjusted the Limited Proteolysis–Mass Spectrometry (LiP–MS) method (Feng et al, 2014; Schopper et al, 2017) for the detection of proteins susceptible to Proteasomal Induced Proteolysis (PiP). In general, LiP–MS uses a non-specific protease applied for a short time to proteome extracts to generate structure-specific proteolytic fingerprints that are subsequently identified by liquid chromatography (LC)-MS. Here, instead of using such a protease, we harnessed the proteolytic activity of the 20S proteasome complex. Comparison of proteolytic fingerprints of proteins in native cell lysates treated with the 20S proteasome relative to proteins in untreated native lysates allowed the detection of putative 20S proteasome substrates and cleavage sites. We then compared the proteolytic fingerprints of lysates from cells grown under oxidative stress to those grown under naïve conditions, and examined the effect that oxidative conditions have on the proteolytic capacity of the 20S proteasome. This strategy has enabled us to identify the repertoire of 20S proteasome substrates, revealing that it is mainly composed of nuclear and stress granule proteins containing IDRs. We also found that the 20S substrates are significantly enriched with RNA and DNA-binding proteins. Moreover, our analysis identified not only fully degraded proteins but also proteins that undergo specific cleavage by the 20S proteasome, at their N- or C-termini, possibly tuning their activity. Our results also demonstrated that oxidized proteasomes have reduced proteolytic activity compared to naïve proteasomes and that oxidized protein substrates display higher structural disorder than naïve substrates. In summary, here we report on an approach that unravels the 20S proteasome substrate landscape, opening up multiple opportunities for investigating the biological cues that influence the complex function.

## Results

### The PiP-MS method identifies 20S substrates within native lysates

To unravel the identity of 20S proteasome substrates in a systematic manner, we established the Proteasome induced Proteolysis coupled-mass Spectrometry (PiP-MS) approach, which is based on the LiP–MS workflow (Feng et al, 2014; Schopper et al, 2017) (Fig. 1A). Briefly, cell lysates are extracted under native conditions. Each sample is then split into two aliquots. The first aliquot undergoes proteolysis by a purified FLAG-tagged 20S proteasome using native conditions (instead of using proteinase K as in LiP–MS). The 20S proteasomes are then removed by FLAG-IP,

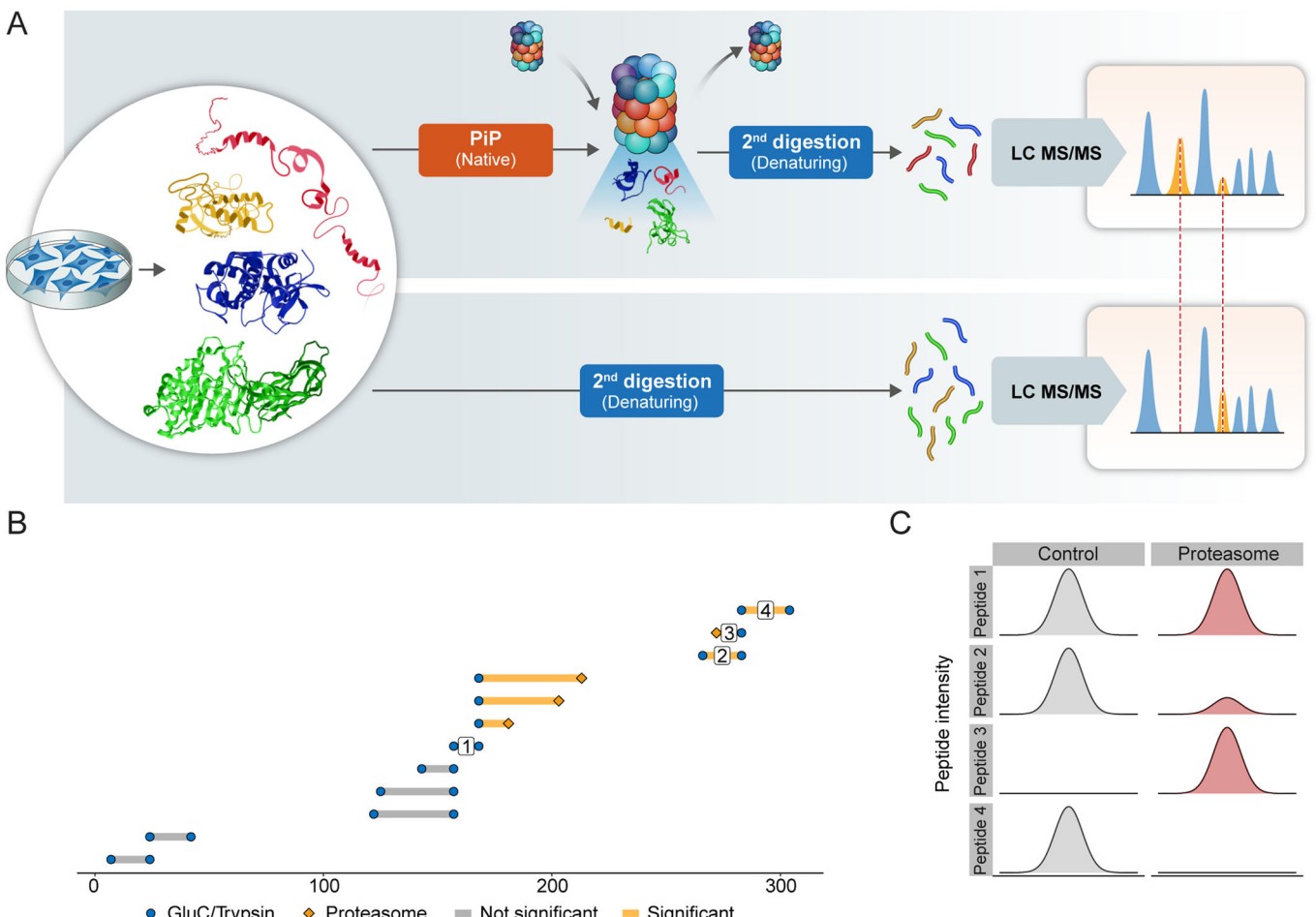

**Figure 1.   Workflow of the PiP-MS methodology for identifying 20S proteasome substrates.**

(A) Proteins were extracted from HEK293T cells under native conditions. Each proteome extract was split into a control sample and a sample subjected to 20S proteasome degradation. Following proteasomal degradation, the 20S complex was depleted from the sample and both control and 20S proteasome degraded samples were denatured and fully digested with GLuC or trypsin to generate peptides amenable to bottom-up proteomics. The peptide mixture was analyzed by LC-coupled tandem MS. (B) Schematic representation of a degraded protein. Each line represents a peptide at a specific amino acid position. The color represents whether the peptide shows significant signs of degradation (orange) or remains stable across the conditions (gray). The blue dots and orange diamonds show whether the peptide has two GluC/trypsin ends or it incorporates some proteasome cleavage. (C) Visual representation of changes in peptide intensity across samples with and without the 20S proteasome. Categories include no significant change (Peptide 1), reduction in the abundance of a specific peptide (Peptide 2), emergence of a new semi-specific peptide (Peptide 3), and complete disappearance of a specific peptide (Peptide 4).

followed by proteome denaturation and simultaneous quenching of residual proteolytic activity, followed by complete digestion with a sequence-specific protease such as GluC or trypsin to generate peptides amenable to bottom-up mass spectrometric analysis. The second (control) sample undergoes incubation without the proteasome, followed by denaturation and sequence-specific digestion (Fig. 1A). Peptide abundances for all detected peptides are then measured in data-independent mode. To identify degraded proteins, the detected peptides of a given protein are compared between the control and proteasome-treated conditions (Fig. 1B,C). The intensity of measured peptides can either remain unchanged (illustrated by Peptide 1 in Fig. 1B,C) or exhibit significant changes between conditions (as observed with Peptides 2–4). A peptide is considered significantly changed if a peptide with two protease-specific ends (specific peptide) decreases in abundance (Peptide 2), a new peptide with a proteasome cleavage site emerges (semi-

specific peptide, Peptide 3), or a specific peptide completely disappears upon the addition of the proteasome. To convert peptide-level information into protein-level information, a strict rule is applied, considering a protein a 20S proteasome substrate if at least 50% of detected peptides show significant change or, in cases where less than 50% show significant change (processed proteins), the significantly changing peptides colocalize in the peptide sequence. To identify 20S proteasome substrates in the human proteome, we applied this pipeline to lysates of HEK293T cells treated with FLAG-tagged 20S proteasomes, immunoprecipitated from the same type of cells. As the second-step digestion enzyme, we used the serine proteinase GluC, which selectively cleaves at the C-terminus of glutamic acid residues (Drapeau et al, 1972). We reasoned that as the GluC cleavage specificity is different from the three catalytic activities of the proteasome, considering that the proteasome caspase-like site

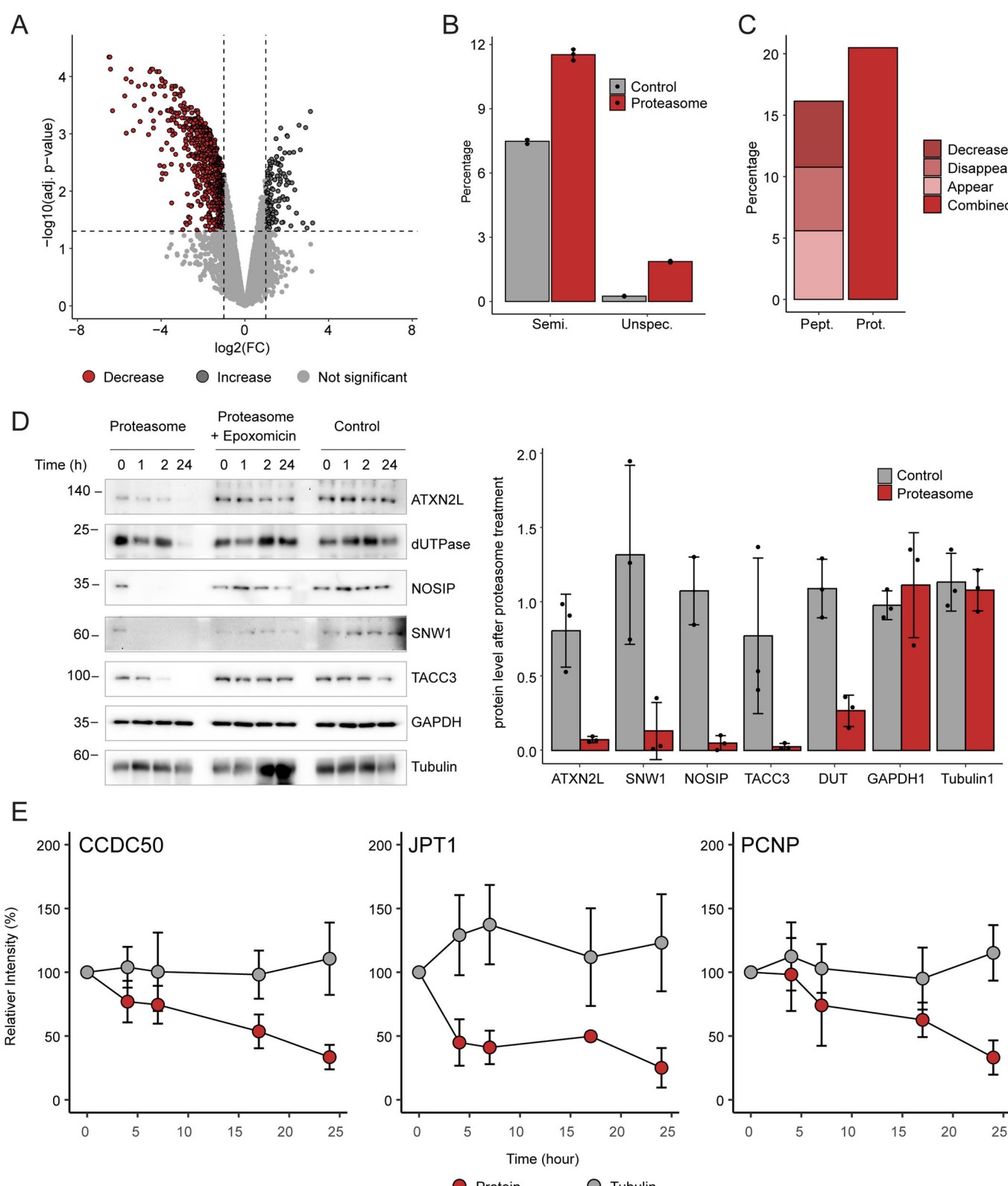

cleaves more efficiently after aspartic acid residues than glutamic acid (Kisselev et al, 2003), we could distinguish proteasome-generated peptides from those produced by GluC. Differential analysis of the peptide abundances from PiP-MS data indicated

significant changes between untreated and 20S proteasome-treated samples (Fig. 2A). We mostly detected a decrease in peptide abundances upon proteasome treatment, indicating degradation of the associated protein regions by the proteasome. Several peptides

**Figure 2.   Detecting 20S proteasome substrates using PiP-MS.**

(A) The volcano plot shows significantly changing peptides in a PiP-MS experiment, followed by a GluC digestion. Each point represents a peptide measured. The color indicates peptides with reduced abundance due to degradation (red, log2 (FC) < −1, adj. $P$ value < 0.05) and increase in peptide abundance (dark gray, log2 (FC) > 1, adj. $P$ value < 0.05) or no significant change (light gray) upon addition of proteasome. $P$ values were calculated using two-sided $t$ test, followed by Benjamini–Hochberg correction, $n = 3$ replicates. Peptides mapping to proteasomal proteins were excluded from the plot and are shown in Fig. EV1A. (B) Percentage of peptides with proteasome cleavage sites; Semi-specific peptides (Semi.,) or unspecific peptides (Unspec., neither of the peptide-ends is GluC specific) in the control (gray) and proteasome (red) condition. Error bars represent the mean $+/-$ SD of $n = 3$ replicates. (C) Percentage of significantly changing peptides or proteins out of all detected peptides and proteins. Peptides are considered significant if the peptide decreases in abundance (Decrease, log2 (FC) < −1), completely disappears from the proteasome samples (Disappear) or if a proteasome-specific peptide appears in the sample. Proteins were considered significant if at least three peptides changed significantly and were not randomly distributed across the sequence (see methods). Values were calculated from a single experiment, with $n = 3$ replicates per condition. (D) Cell lysate were treated with purified 20S proteasomes in the presence or absence of the irreversible proteasome inhibitor epoxomicin. Samples were analyzed by western blot using antibodies against the five selected targets: ATXN2L, DUTPase, NOSIP, SNW1and TACC3. Anti-GAPDH and tubulin antibodies were used as control. Quantification of three independent experiments after 24 h incubation with the 20S proteasome is displayed on the right, error bars represent SEM. (E) Cells were treated with the ubiquitination inhibitor, TAK-243, together with cycloheximide, in order to inhibit protein synthesis, and cells were harvested after 4, 7, 17, and 24 h. Changes in the levels of the CCDC50, JPT1 and PCNP, relative to the initial time point were quantified from four independent experiments. Error bars represent SEM. Source data are available online for this figure.

that increased in abundance in proteasome-treated samples originated from subunits of the 20S proteasome itself or from proteasome-associated proteins that we likely added to the samples as contaminants of the spiked immunoprecipitated FLAG proteasomes (EV1A,B; Dataset EV1).

In addition, we observed a substantial increase in the number of semi-specific (i.e., with only one GluC specific terminus) and unspecific peptides (i.e., with neither GluC specific terminus) in 20S proteasome-treated samples (Fig. 2B), indicating proteasomal dependent degradation and pinpointing proteasome cleavage sites. Overall, for 20% of proteins identified in the proteomic analysis, we detected at least three peptides indicative of 20S proteasome cleavage (Fig. 2C). Altered peptides from this set of proteins were either peptides that decreased in intensity (adj. $P$ value < 0.01, log2 (FC) < −1), completely disappeared or new semi-specific peptides that appeared upon proteasome treatment (Fig. 2C). Overall, we identified 280 candidate substrates of the 20S proteasome from 2180 peptides pinpointing proteasome-cleaved regions.

To validate that the PiP-MS workflow indeed identifies 20S proteasome substrates we performed two orthogonal experiments. We selected five candidate substrates from our PiP-MS dataset (Fig. EV1C) and monitored their susceptibility to 20S proteasome-mediated degradation by time-dependent degradation assays. In these assays, cellular lysates were incubated with purified human 20S proteasome for different time periods, followed by western blot analysis with an antibody against the putative substrate to evaluate its potential degradation (Fig. 2D). We focused on ATXN2L (113.4 kDa), involved in stress granule and P-body formation, the nucleotide metabolism enzyme, DUTPase (17.7 kDa), the E3 ubiquitin-protein ligase NOSIP (33.2 kDa), the spliceosome component SNW1 (61.5 kDa), and the protein TACC3 (90.4 kDa), involved in cell growth and differentiation. Western blot analyses confirmed that all five proteins are indeed degradation targets of the 20S proteasome. Moreover, the proteasome inhibitor epoxomicin protected the selected proteins from degradation, while the two control proteins GAPDH and tubulin, which are not known to be 20S proteasome substrates and were not identified by our screen, were resistant to degradation.

We next examined whether our observations from cell lysates are also relevant within the context of the cellular environment. To this end, we performed a cycloheximide (CHX) chase assay, which blocks protein synthesis, to monitor the cellular stability of the PiP-

MS candidates. Taking into account that in cells, identified substrates might be sent for degradation via both the 20S and 26S proteasomes (Ben-Nissan and Sharon, 2014a), we also assessed whether the stability of the proteins was associated with the 20S or the 26S proteasome degradation route. We therefore treated cells with the ubiquitination inhibitor TAK-243 (Hyer et al, 2018), which prevents ubiquitin-dependent degradation by the 26S proteasome, and monitored, over time, the decay in the level of three candidate 20S proteasome targets CCDC50 (35.8 kDa), a signaling protein; the cell cycle and cell adhesion regulator JPT1 (16.0 kDa); and the nuclear protein PCNP (18.9 kDa) (Fig. EV1D). We observed that these three proteins were degraded in an ubiquitin-independent manner in the presence of TAK-243 (Figs. 2E and EV2). These results imply that the proteins identified as PiP-MS hits are indeed substrates of the 20S proteasome. While analyzing the generated peptides, we noticed that 82% of the GluC unspecific peptides had no tryptic end, and 16% of peptides in this group were semi-tryptic (i.e., cleaved at only one end by trypsin-like protease activity) (Fig. EV3A). This result emphasizes that tryptic-peptides generated by the 20S proteasome are scarce, in accordance with previous observations (Wolf-Levy et al, 2018). We therefore decided to use trypsin as sequence-specific protease in our denaturing digestion step instead of GluC, in all subsequent experiments. This protocol optimization enabled us to substantially increase the number of peptides and proteins that were identified by more than threefold (Fig. EV3B,C).

## Oxidized 20S proteasomes are less active than the naïve complexes

The 20S proteasome displays increased resistance to oxidative stress in comparison to the 26S proteasome complex (Reinheckel et al, 1998). In addition, under these conditions, there is initial disassembly of the 26S proteasome into its 20S and 19S components, followed by de novo synthesis of 20S proteasome subunits (Pickering and Davies, 2012a; Pickering et al, 2010; Seifert et al, 2010), increasing the total amount of free 20S proteasome in the cell. However, how the 20S proteasome substrate landscape changes under conditions of oxidative stress and whether the proteolytic capacity of the oxidized 20S proteasome is different from that of the naïve complex is not known. Therefore, after establishing the PiP-MS workflow we examined the impact of

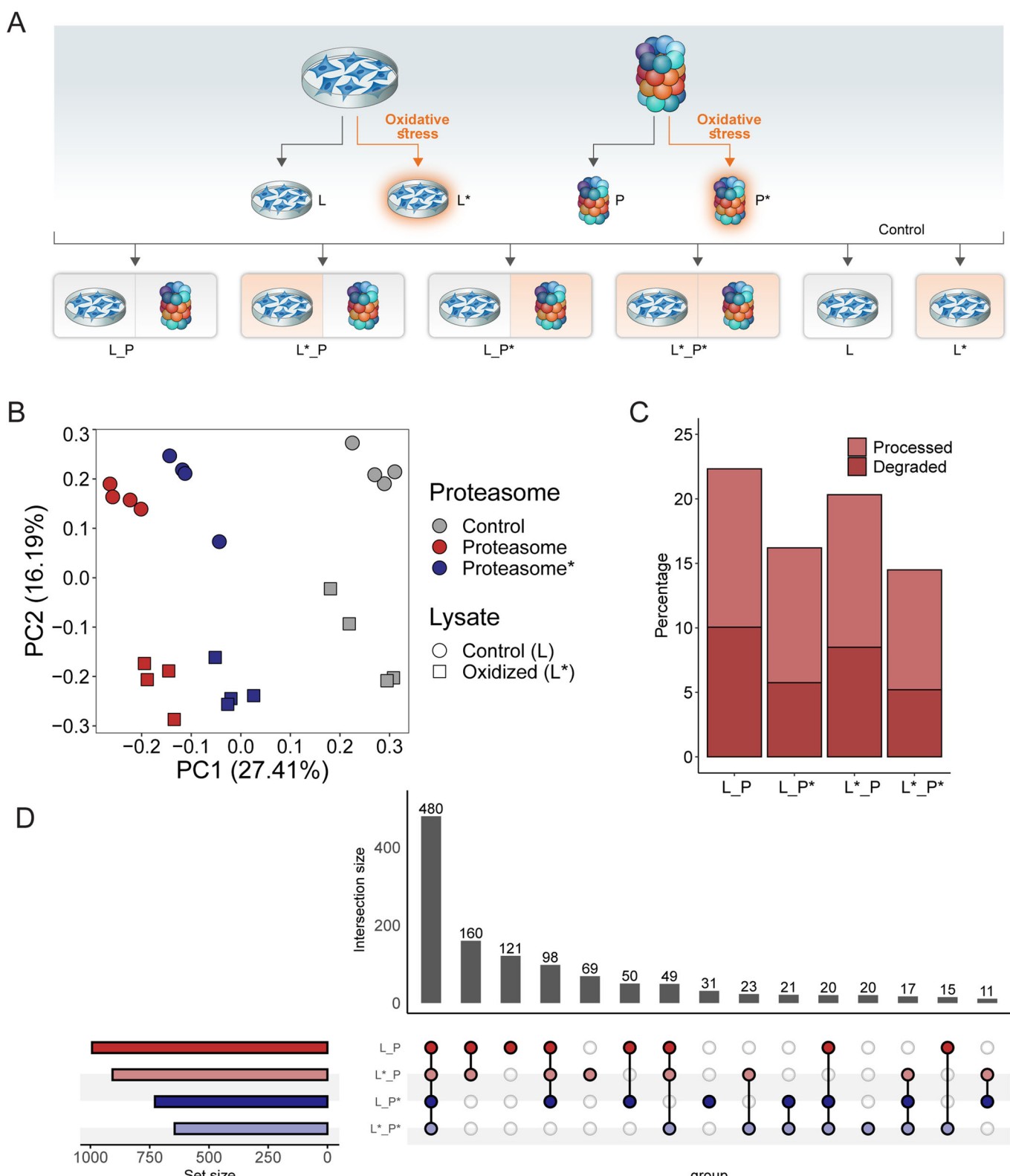

oxidative stress on both the proteome and the 20S proteasome and repeated the PiP-MS analyses under four experimental conditions. We used two types of cell lysates, one from untreated cells and one from cells subjected to oxidative stress, and two types of purified 20S proteasomes, one isolated from untreated cells and one from cells subjected to oxidative stress. From these, we generated four different types of proteasome-treated samples that we compared to untreated or oxidized lysates without added proteasomes (Fig. 3A).

**Figure 3.   More proteins are cleaved by the naïve 20S proteasome then its oxidized counterpart.**

(A) Overview of the experimental pipeline. Lysates were prepared from cells subjected to oxidative stress (L*) and those grown under normal conditions (L). Similarly, human 20S proteasomes were isolated from cells exposed to oxidative stress (P*) and naïve cells (P). Four different sample conditions were prepared by combining the treated and untreated 20S proteasome and lysates. The two lysate samples, naïve (L) vs. oxidized (L*), without the proteasome, served as control. (B) Principle component analysis of the PiP-MS experiment. Each point represents a run (6 different conditions, four replicates each). The colors indicate whether proteasome was added, and the shape indicates whether the control or oxidized lysate was used. (C) Percentage of significantly degraded proteins out of all detected proteins for the four conditions. In case more than 75% of all detected peptides in a protein were significantly changing, the protein was considered completely degraded. Otherwise, the protein was considered processed. Values were calculated from a single experiment with $n = 4$ replicates for each condition. (D) Upset plot indicating the overlaps of proteasomal targets across different conditions. The bars indicate the intersection size and the points indicate the conditions included in the intersection. Values were calculated from a single experiment with $n = 4$ replicates for each condition.

Oxidative stress was induced by exposing cells to diethylmaleate (DEM), a compound that depletes glutathione, leading to the accumulation of reactive oxygen species within the cell (Kalo et al, 2012). The generation of oxidative stress was confirmed by the upregulation of the two antioxidant enzymes NQO1 and NQO2, and the 20S proteasome subunit PSMA1 (Fig. EV4) (Rashid et al, 2021).

The six different conditions with four replicates per sample were analyzed by data-independent acquisition MS. Principal component analysis (PCA) of the peptide abundances across all the conditions showed a clear separation between naïve and oxidized proteasome states based on the first component, while the second component separated the samples based on the lysate condition, naïve versus oxidized (Fig. 3B). PCA analysis indicated that samples containing oxidized proteasome were more similar to the control samples (i.e., no proteasome added) than the naïve proteasome, suggesting that the oxidized proteasome is less active than the naïve complex. This view is strengthened by the fact that there was about 25% reduction in the fraction of degraded proteins when oxidized 20S proteasomes were used (Fig. 3C). The state of the lysate, however, had a smaller effect on the number of degraded proteins. Analysis of the overlapping hits among the four different samples, i.e., the combinations of naïve and oxidized proteasome and lysate, showed a high number (480) of overlapping targets between all conditions (Fig. 3D; Datasets EV2 and EV3), with an additional 120 targets shared between conditions with naïve proteasome. This shows that the majority of proteasomal targets do not depend on the oxidation status of the lysate or the proteasome. Nevertheless, it is clear that the oxidation status of the 20S proteasome affects the proteolytic activity more significantly than the state of the lysate, with the oxidized 20S proteasome displaying an overall decrease in degradation capacity. These properties are likely very important under oxidative cellular conditions, in which there is an increased number of 20S particles available and a need to tune its function.

To further disentangle changes in PiP-MS peptide intensities due to proteasome oxidation, we compared the hits observed after adding the different types of proteasomes, naïve versus oxidized, to naïve or oxidized lysates (Fig. 4A). The strong increase in degradation when the naïve 20S proteasome was used supports the view that this complex is more active than its oxidized counterpart. To strengthen this observation, we decided to focus on a specific proteasome substrate, α-synuclein. We purified both naïve and oxidized proteasomes and examined their proteolytic capacity using an activity-based probe, MV151 (Verdoes et al, 2006), which labels the three catalytically active β-subunits (Fig. 4B). No difference was detected in band intensities of the

distinct subunits between the two proteasomes, suggesting that oxidation does not perturb the active sites. Nevertheless, the time-dependent degradation assays indicated that the oxidized proteasome had reduced capacity of degrading its known target, α-synuclein (Figs. 4C and EV5), supporting our PiP-MS results (Fig. 4D). Considering that both types of proteasomes reacted similarly with the activity-based probe that binds to the active sites, these results suggest that the proteasome modifications induced by oxidative conditions do not disturb the enzymatic sites, rather they may trigger a structural or allosteric perturbation that affects degradation capacity.

Oxidative stress leads to a variety of reversible and irreversible post-translational modifications of proteins, including carbonylation, and modifications of cysteine, methionine and tyrosine residues (Cai and Yan, 2013). These chemical modifications have the potential to disrupt protein structure and facilitate partial unfolding, subsequently rendering oxidatively damaged proteins to degradation by the 20S proteasome. Therefore, we wished to examine whether the structural properties of PiP-MS hits of the oxidized lysate are different from those of the naïve lysate. Namely, we asked whether oxidative stress leads to 20S proteasome-mediated degradation of proteins that under basal conditions are folded and not susceptible to degradation. We therefore compared the average disorder, predicted by DisoPred, of PiP targets and of all detected proteins. The average disorder of hits from the oxidized lysate was significantly higher compared to that of hits from the naïve lysate (Fig. 4E). This result suggests that loss of structure, due to oxidative damage, is not the main factor that sensitizes proteins towards degradation by the standard 20S proteasome.

## The majority of 20S proteasome substrates are disordered RNA and DNA-binding proteins

We next asked whether we could find common properties amongst the identified 20S proteasome substrates. To this end, we calculated various protein features associated with either protein sequence or structure and checked whether significant differences could be observed between the set of degraded proteins and all detected proteins. For all four examined conditions, we found that the candidate 20S proteasome substrates are enriched in structural disorder, a feature that is known to sensitize proteins towards 20S proteasome-mediated degradation (Fig. 5A) (Ben-Nissan and Sharon, 2014a; Kumar Deshmukh et al, 2019). Moreover, significant enrichment of molecular recognition features (MoRFs) is detected. These are defined as short protein-binding regions that undergo disorder-to-order transitions (induced folding) upon

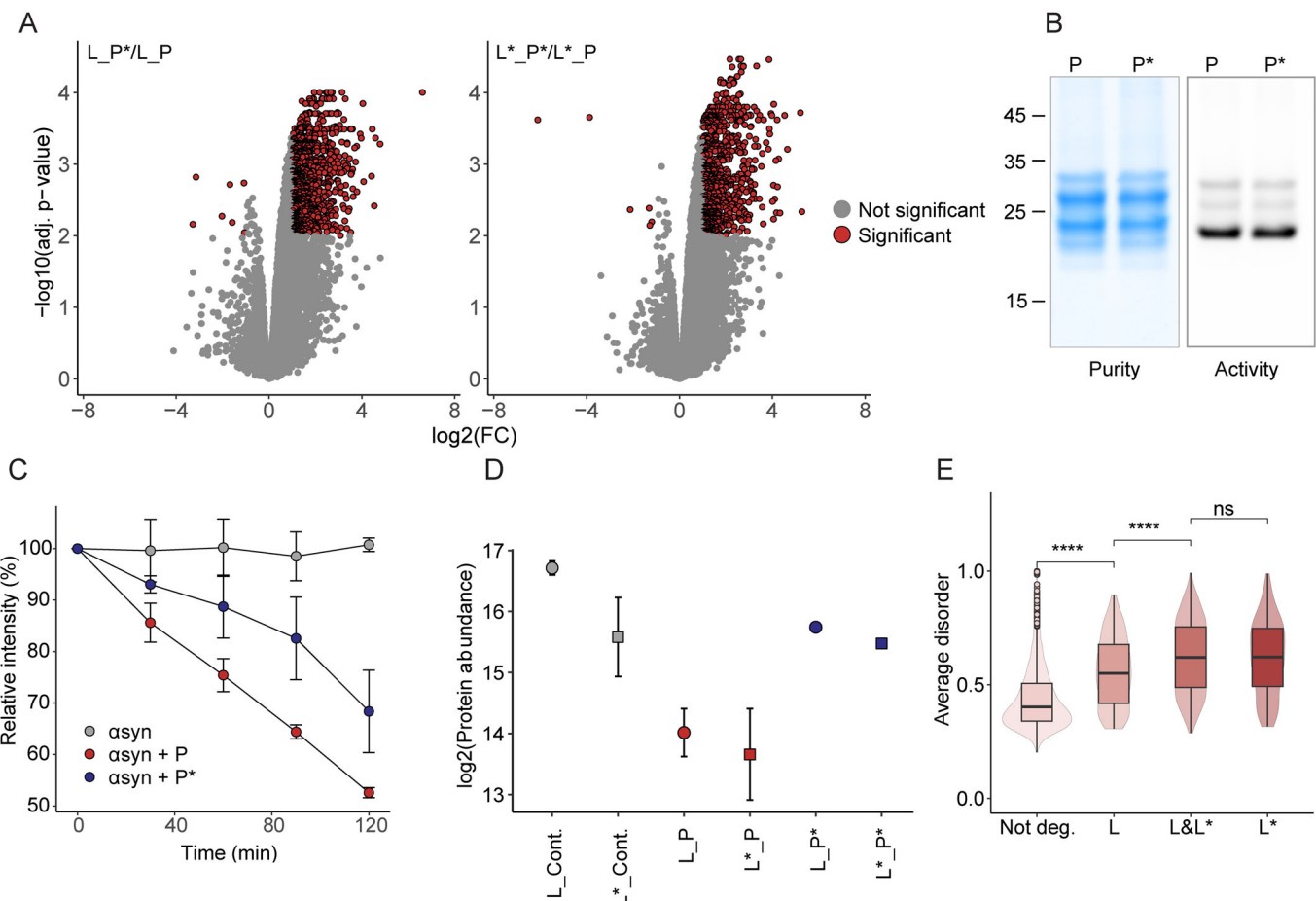

**Figure 4. The degradation capacity of oxidized proteasome is reduced compared to the naïve proteasome.**

(A) Volcano plot shows peptides digested significantly different when an oxidized proteasome (P*) is used instead of the naïve proteasome (P) for digestion of naïve lysate (L, left) and oxidized lysate (L*, right). Each point represents a peptide measured. The color indicates significant change (red, abs (log2 (FC)) > 1 & adj. P value < 0.01). Increased FC indicates that the peptide is digested more when naïve proteasome is used. P values were calculated using two-sided t test, followed by Benjamini–Hochberg correction, n = 4 replicates. (B) Validation of the purity and activity of the naïve and oxidized 20S proteasome complexes that were added to the lysate. The purity of the complexes was assessed by SDS-PAGE analysis followed by coomassie staining, and the activity of the 20S proteasomes was analyzed by incubating proteasomes with the MV151 activity-based probe before separation by SDS-PAGE. (C) Quantification of time-dependent degradation assays of α-synuclein (αsyn) in the presence of naïve (P) and oxidized (P*) 20S proteasomes. The graph represents the averages of three independent experiments with SD. (D) PiP-MS results of α-synuclein degradation in different conditions. Values were calculated from a single experiment with n = 4 replicates for each condition. (E) Average disorder of proteins that are not degraded (Not deg-.), are degraded only in control lysate (L), only in oxidized lysate (L*) or in both lysates (L&L*). Values were calculated from a single experiment with n = 4 replicates for each condition. Significance is determined using two-sided Wilcoxon test (ns = P value > 0.05, ****P value < 0.0001). Horizontal lines define the median and boxes the 25th and 75th percentiles; whiskers represent the maximum and minimum values. Source data are available online for this figure.

binding to protein partners, substrates or ligands (Katuwawala et al, 2019). This may indicate that the substrates in question lacked a binding partner at the time of experimentation, resulting in their disordered state and heightened susceptibility toward degradation. They are also generally lower in aliphatic index and hydrophobicity, which is in accordance with their enrichment in disordered regions. We also found that the 20S proteasome substrates have a higher content of proline, glutamic acid, arginine, glutamine, serine and alanine, and on average less hydrophobic amino acids such as isoleucine, phenylalanine, valine, tyrosine, leucine, tryptophan and cysteine (Fig. 5B). The latter are generally found at the core of a globular protein, while the former are known as hallmarks of proteins containing IDRs (Zhao and Kurgan, 2022). Taken together, this suggests that the lack of structure is a distinguishing feature of 20S proteasome substrates.

Next, we evaluated the functional properties of the identified 20S proteasome substrates. We started by examining the cellular localization of these proteins. Interestingly, we found that most of the substrates were localized to the nucleus and to stress granules (Fig. 5C). Membrane proteins, on the other hand, were excluded from this group, suggesting a safeguarding effect of the membrane in protecting proteins from degradation. Gene Ontology (GO) analysis based on molecular function and biological process enrichment revealed that the most enriched terms are RNA- and DNA-binding proteins (Fig. 5D). This observation likely explains the nuclear localization of the majority of the substrates. Other interesting functions included cadherin-binding proteins and proteins containing leucine zipper domains, which are both unstructured in the absence of a binding partner (Shapiro and Weis, 2009; Vancraenenbroeck and Hofmann, 2018), in line with

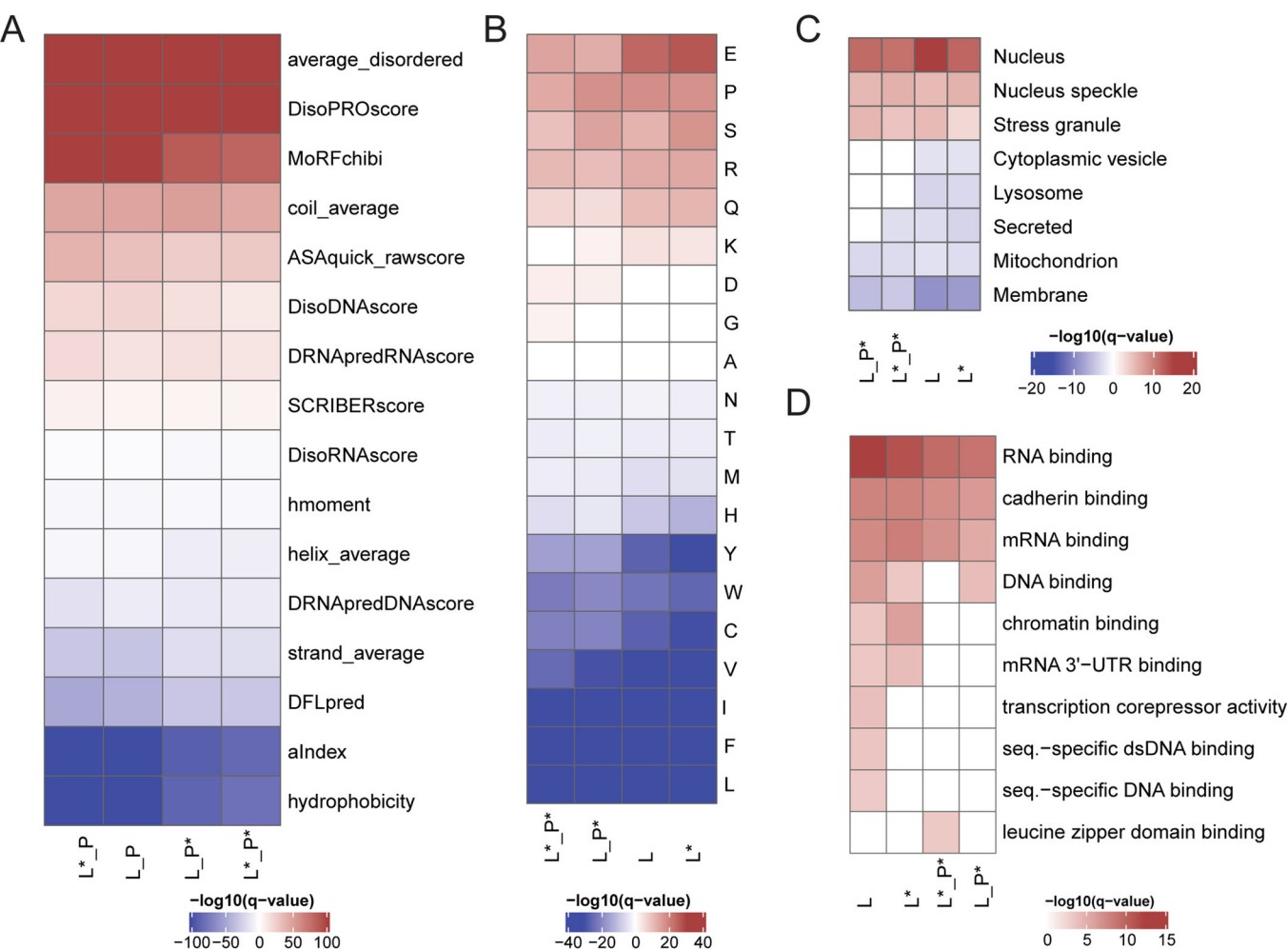

**Figure 5. 20S proteasome substrates are enriched with DNA- and RNA-binding proteins.**

(A) The heatmap shows the most significantly different features (adj. *P* value < 0.0001 in at least one condition) between degraded proteins vs. all detected proteins. The color intensity indicates significance levels. The color indicates whether the feature is higher (red) or lower (blue) in degraded proteins compared to the rest. Feature significance was determined by *t* test followed by correction for multiple hypothesis testing with the Benjamini–Hochberg method. Average secondary structure prediction (strand_average, coil_average, and helix average) and average disorder values were predicted using DISOPRED3 (Jones and Cozzetto, 2015). Hydrophobicity, hmoment, and aliphatic index (aIndex) were calculated from protein sequence. The propensity to bind proteins (SCRIBER) (Zhang and Kurgan, 2019), to bind DNA (DRNApredDNA) (Yan and Kurgan, 2017) or RNA (DRNApredRNA) (Yan and Kurgan, 2017), DisoRNA for RNA binding to disordered regions and DisoPRO for proteins binding to disordered regions (Zhao et al, 2021a), and the presence of molecular recognition features (MoRFchibi) (Disfani et al, 2012), presence of disordered flexible linker (DFLpred) (Meng and Kurgan, 2016) and accessible surface area (ASA) (Zhao et al, 2021a) were downloaded from the databases (Zhao et al, 2021a). (B) The heatmap shows amino acids that are enriched (red) or significantly underrepresented (blue) in degraded proteins. The color intensity indicates significance levels. (C) Molecular-function GO-enrichment for significantly degraded proteins. (D) Localization enrichment for significantly degraded proteins.

the fact that proteins containing IDRs are frequently involved in mediating interactions.

To gain more insights into the mechanism of proteasomal degradation, we investigated whether candidate 20S substrates are generally completely degraded or processed at specific sites. For each candidate substrate, we calculated the degradation percentage as the fraction of peptides that were significantly degraded upon the addition of proteasome out of all detected peptides for that protein. We then compared the distribution of degradation percentage across all conditions (Fig. 6A). Here again we noticed that under oxidative conditions, proteasomal degradation was less efficient. While many proteins were completely degraded upon the addition of naïve proteasome (degradation percentage = 100%), most

proteins did not undergo complete degradation upon the addition of oxidized proteasome. To investigate whether we could identify determinants of complete or partial cleavage of a protein, we grouped proteins based on their degradation percentage in the naïve lysate and proteasome condition (L_P) and asked whether significantly different characteristics could be observed for different groups. We observed that shorter proteins were more frequently fully degraded (Fig. 6B). Moreover, we found that the extent of degradation was correlated with the score on SCRIBER, which measures protein-protein interaction domains (Zhang and Kurgan, 2019), and MoRFchibi, which measures the density of molecular recognition features (Disfani et al, 2012). Based on our data, an increase in either of these two scores proportionally leads to

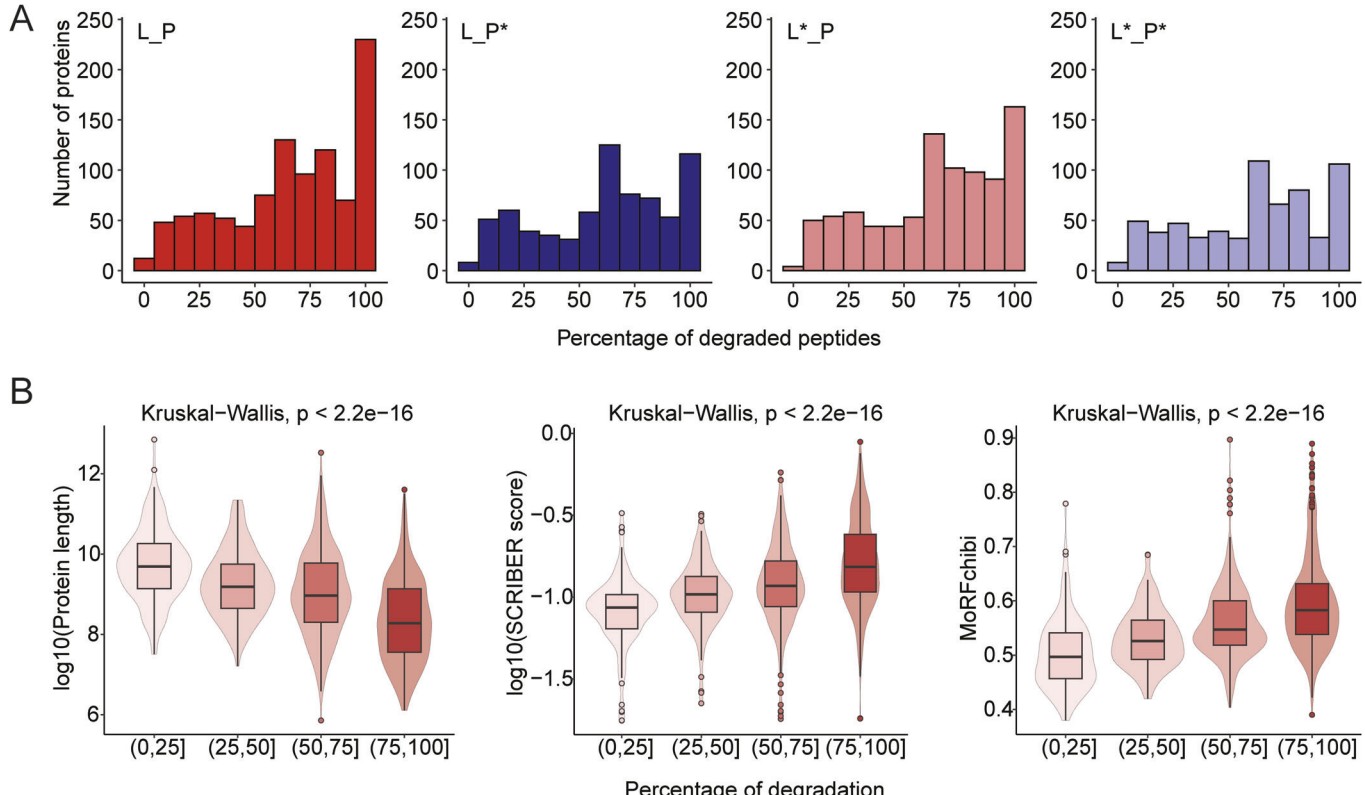

**Figure 6. The extent of protein degradation by the 20S proteasome is correlated with the presence of protein-protein interaction domains and disorder-to-order motifs.**

(A) Distribution of degradation percentage (degraded peptides/all peptides) for all detected proteins with at least three detected peptides, across all conditions. The values were calculated based on a single experiment, with $n = 4$ replicates per condition. (B) Distribution of protein length, SCRIBER score (propensity for protein binding) (Zhang and Kurgan, 2019) and presence of molecular recognition features (MoRFchibi) (Disfani et al, 2012) for proteins that are processed to a different extent, as calculated in (A). Values were calculated from a single experiment with $n = 4$ replicates for each condition. Significance is determined using Kruskal–Wallis test. Horizontal lines define the median and boxes the 25th and 75th percentiles; whiskers represent the maximum and minimum values.

enhanced 20S proteolysis, possibly suggesting that proteins that are lacking their inherent binding partners are more likely to be completely degraded.

## Proteins are proteolytically processed by the 20S proteasome

The observation that in some proteins we detect the selective cleavage of specific regions of the polypeptide is in agreement with previous knowledge on 20S activity. It was shown that there are proteins in which the 20S proteasome cleaves at specific sites to generate functional cleavage products, rather than degrade them to completion (Olshina et al, 2018). For example, p53, NF-kappaB1, Hsp70, eIF4F and eIF3 were shown to undergo selective proteolysis of a specific disordered region within the polypeptide chain (Baugh and Pilipenko, 2004; Baugh et al, 2009; Moorthy et al, 2006; Morozov et al, 2017; Solomon et al, 2017). This process influences diverse cellular pathways such as transcription, protein synthesis, and the response to cellular stress (Olshina et al, 2018). Thus, we assessed whether we could identify protein substrates in which only peptides in the N- or C-termini had significantly lower abundance upon the addition of the 20S proteasome, while all the remaining peptides remained unchanged.

We found that for a large fraction of the candidate proteasome substrates, a significant change was detected for all peptides along the protein sequence, indicating that these are degraded to completion (Fig. 6A). However, for 327 proteins, we could find evidence of selective degradation at specific protein regions. We next asked whether the position of the cleaved peptide was more towards the C- or N-terminus, compared to nonchanging peptides. We then calculated the p value individually for each protein with at least three significantly changing and three unchanged peptides. This enabled us to identify ~240 proteins that were significantly cleaved at their N- or C-terminus, ~70 proteins that are cleaved at both ends and 20 proteins that are cleaved in the middle of their sequence (Fig. 7A; Dataset EV4). Analysis of the structural properties of the selectively cleaved proteins revealed a clear pattern wherein folded regions exhibited resilience against degradation, whereas intrinsically disordered regions (IDRs) were subject to cleavage by the 20S proteasome (Fig. 7B).

To validate the identification of partially cleaved proteins by PiP-MS we carried out in vitro degradation assays. Specifically, we selected three substrates, each displaying a different cleavage region: PCPB2, an oncogenic splicing factor that is cleaved at its C-terminus; SF3A1, a subunit of the splicing factor 3a protein complex, which is cleaved in the middle of the polypeptide; and

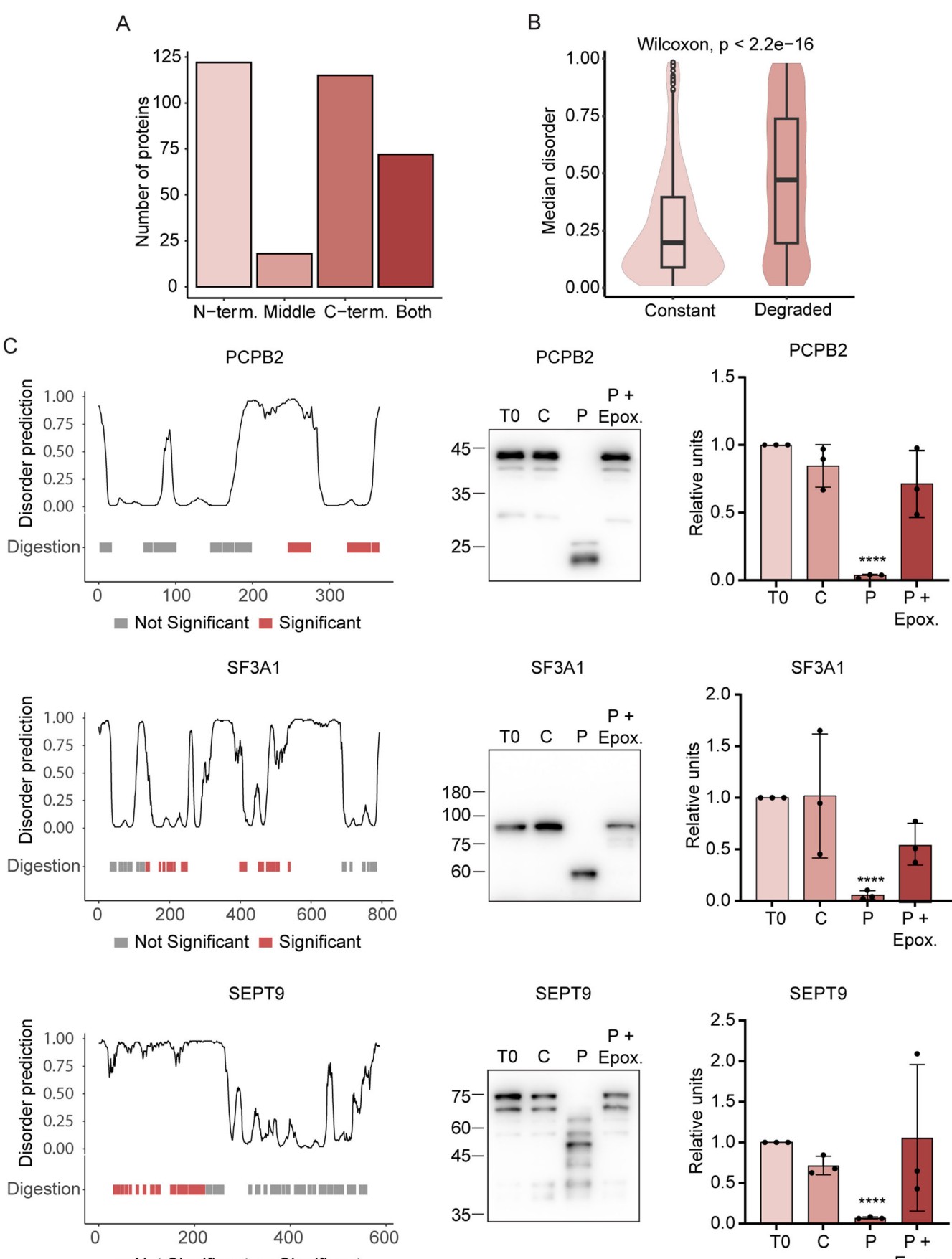

**Figure 7.  Proteolytic processing of substrates by the 20S proteasome.**

(A) Number of processed proteins for control lysate (L) and control proteasome (P) that are digested from the N-terminal, C-terminal, middle or both ends. (B) Distribution of median disorder of protein regions from 327 processed proteins for regions that show significant degradation (Degraded) and regions that show constant abundance (Constant). Horizontal lines define the median and boxes the 25th and 75th percentiles; whiskers represent the maximum and minimum values. The significance level was determined using two-sided Wilcoxon test. (C) The plots show three example of proteins: PCPB2, SF3A1, and SEPT9,  which are cleaved differently by the 20S proteasome, at the C-terminal, middle, and N-terminal end of the polypeptide chain, respectively. The plot on the bottom left indicates the significantly degraded peptides (red) among all detected peptides (gray) across the protein sequence. The top left plot shows the disorder prediction across protein sequence, where the high probability indicates disordered regions. The middle image shows lysate degradation assays in the absence (C) or presence of the 20S proteasome (P), with and without the proteasome inhibitor epoxomicin (Epox). Averaged quantification of three independent biological experiments is displayed on the right. Statistical analysis was calculated for the T0 and P samples using unpaired two tailed t test. P value (****) < 0.0001. Error bars represent SEM. Source data are available online for this figure.

SEPT9, a member of the septin family that is cleaved at its N-terminus. Degradation was assessed by western blot analysis visualized using monoclonal antibodies recognizing the regions that were identified by PiP-MS to remain unaltered upon addition of the proteasome, i.e., regions that are not degraded. As shown in Fig. 7C, PCBP2, SF3A1 and SEPT9 were not entirely degraded, but rather processed to a smaller product. Moreover, the apparent size of the processed product was in agreement not only with that expected from the PiP-MS data but also with the protein's intrinsic disorder prediction along its sequence. This suggests that only a disordered fragment underwent removal, while the structured regions served as termination signals for 20S proteasomal processing. Taken together, these results suggest that the 20S proteasome may facilitate the post-translational processing of proteins, which may lead to the modulation of protein function and alteration of downstream processes.

## Discussion

Emerging evidence highlights the involvement of 20S-mediated degradation in key cellular processes occurring in both the cytoplasm and nucleus. These activities encompass neuronal stimulation (Ramachandran et al, 2018; Ramachandran and Margolis, 2017), antigenic peptide production (Dalet et al, 2010; Vigneron et al, 2019), oxidative stress (Aiken et al, 2011; Lefaki et al, 2017a; Raynes et al, 2016b), hypoxia (Sahu et al, 2019) and post-translational processing (Baugh and Pilipenko, 2004; Moorthy et al, 2006; Olshina et al, 2018; Solomon et al, 2017; Sorokin et al, 2005). Moreover, it was shown that cleavage by the 20S proteasome is not a random phenomenon, but rather a tightly regulated biological process (Deshmukh et al, 2023; Olshina et al, 2020). Nevertheless, a complete understanding of the 20S proteasome activity cannot be contemplated without unraveling the repertoire of substrates degraded by this complex, and how these and the activity of 20S proteasomes are altered under stressful conditions. Here, we present a systematic investigation of 20S proteasome substrates in native human cell lysates. The analysis revealed that 20% of the proteomic analysis identified proteins undergo proteolysis by the complex. We discovered proteins that are degraded to completion and others that appear to undergo specific proteolytic processing at a specific disordered region, thus providing a valuable resource for the field of proteasomal degradation. In addition, we mapped the substrates that undergo proteolysis under basal and oxidative conditions and compared the functionality of naïve and oxidized 20S proteasome complexes. Overall, we discovered that although structural disorder and tendency to undergo disorder-to-order transitions are the main criteria for being a 20S substrate, cellular

conditions also influence the protein susceptibility towards 20S proteasome degradation.

In humans, 44% of protein-coding genes are predicted to contain disordered segments of >30 amino acids in length (van der Lee et al, 2014). Regions that lack a well-defined three-dimensional structure represent a major functional advantage, as they enable proteins to interact with a broad range of substrates with relatively high specificity and low-affinity. Intrinsically disordered proteins (IDPs) or proteins with IDRs often undergo a disorder-to-order transition upon interacting with protein partners or binding to substrates, nucleic acids and ligands (Deiana et al, 2019; Yan et al, 2016). Proteins from this class are not homogeneously distributed in the cell and are rather localized to several subcellular compartments, such as the nucleus and its membrane, cytoskeleton, centrosome, and cytoplasm (Zhao et al, 2021b). The diverse array of biological functions of proteins with IDRs includes roles in signaling, transcription, transcriptional regulation, translation, spermatogenesis and DNA condensation (Babu, 2016; Bondos et al, 2021). The IDPs that are efficiently degraded by the 20S proteasome are enriched in nuclear proteins that bind DNA and RNA. This observation is supported by the fact that the abundance of DNA and RNA-binding proteins is tightly regulated. Increased levels of DNA or RNA-binding proteins changes the rate constant by which they associate with their nucleic acid target sites, permitting binding to low-affinity sites that are not occupied at physiological protein concentrations (Muller-McNicoll et al, 2019). Therefore, excessive levels of DNA and RNA proteins can induce binding to "non-physiological" targets, leading to neomorphic activity. This scenario could explain the rapid need to eliminate the unbound form of these proteins, as potentially facilitated by the 20S proteasome in this context. However, considering that subtle alterations in cellular conditions like ionic salt concentration, pH, ATP levels, or post-translational modifications can impact disorder-to-order transitions, the 20S substrate landscape is likely to be dynamic and contingent on the cellular state.

Previously, it was shown by experiments with purified proteins that proteins like hemoglobin (Grune et al, 2003), calmodulin (Ferrington et al, 2001) and ferritin (Shringarpure et al, 2003) that are not susceptible to degradation by the 20S proteasome under basal conditions, become 20S substrates following oxidative stress. Based on these findings, it was suggested that the loss of structure due to oxidative damage sensitizes globular, soluble proteins with defined structures to 20S proteasome-mediated degradation (Grune et al, 2003). Our PiP-MS data, however, demonstrated that proteins susceptible to 20S proteasome degradation following oxidative stress are already proteins with a high degree of disorder, and not folded proteins that were damaged. This contrasting observation may be due to the nature of our approach, which yields a global view, as opposed to

previous individual case studies. Moreover, here we used the standard 20S proteasome, which was previously found to have a reduced ability to degrade oxidized proteins in comparison to the immuno- and intermediate 20S proteasomes (i.e., complexes that contain a mixture of standard and inducible proteasome subunits) (Abi Habib et al, 2020). Particularly, it was shown that the chymotrypsin-like immuno subunit, PSMB8, plays a critical role in the degradation of oxidized proteins (Abi Habib et al, 2020). Moreover, it was discovered that the ATP-independent proteasome regulators, PA28αβ and PA28γ also take part in the degradation of oxidized proteins (Pickering and Davies, 2012b). These studies indicated that in response to oxidative stress, there is not only an increase in expression of both PA28αβ and PA28γ over the subsequent 24 h, but also enhanced capacity of the 20S proteasome to selectively remove oxidative damaged proteins (Pickering and Davies, 2012b). Similarly, Hsp70 was suggested to mediate the shuttling of oxidized proteins to the 20S proteasome (Reeg et al, 2016). Thus, further studies will be required to establish whether proteasomes containing the PSMB8 subunit and factors like PA28αβ, PA28γ and Hsp70 are required for sensitizing folded proteins exposed to oxidative damaged to 20S proteasome-mediated degradation.

Exposure to reactive oxygen species does not only affect cellular proteins but also the proteasome itself (Hohn and Grune, 2014). Specific 20S proteasome subunits have been shown to acquire post-translational modifications like glutathionylation, carbonylation, glycoxidation and lipoxidation upon oxidative stress, impairing the proteasome activity (Lefaki et al, 2017b). As we show here, the three catalytic sites of the complex are not perturbed due to oxidative modifications, retaining the ability to bind to the MV151 activity probe, suggesting that alterations in the gate conformation and/or the propagation of allosteric structural changes are those that impair the complex function. The decline in proteasome activity can have a protective cellular effect, since immediately following an oxidative insult, disassociation of the 19S regulator from the 26S proteasome complex occurs, leading to a large increase in 20S cellular concentration (Raynes et al, 2016a). To prevent rampant degradation, tuning down the proteolytic activity of the 20S complex might be beneficial. In the later phase, the response to oxidative stress involves Nrf2-dependent transcriptional activation that leads to the expression of 20S proteasome subunits (Pickering et al, 2012), consequently generating new 20S complexes that are fully functional, unlike the aberrant oxidized complexes. Taken together, the data further indicates the complexity of 20S regulation through a variety of mechanisms that coordinate its cellular activity.

In the early 1990s, it was discovered that the activity of the 20S proteasome is not restricted to complete degradation of its protein substrates, but that the complex can cleave at specific sites to generate functional cleavage products (Fan and Maniatis, 1991). This exciting finding identified the processing of p105, the precursor of the p50 subunit of the mammalian transcription factor NF-κB showing that the complex can activate dormant signaling molecules (Fan and Maniatis, 1991; Rape and Jentsch, 2002). Since then additional specific examples have been revealed, such as cleavage of the translation initiation factors eIF4F and eIF3 that leads to assembly inhibition of the of the pre-initiation complex (Baugh and Pilipenko, 2004) and cleavage of the tumor suppressor p53 precisely at position 40, generating the dominant negative Δ40p53 isoform (Solomon et al, 2017). The approach we describe here is directly transferable for a comprehensive identification of such proteasomally processed proteins, as

it is performed under native conditions, conserving the proteins three-dimensional structure. This enables identification of cleaved and unaltered protein regions due to structural constraints. Using PiP-MS we discovered that 20S proteasomal processing is a widespread phenomenon, with more than 200 proteins partially processed by the complex. Our results also indicate that the cleavage is not restricted to the N- or C-terminus of proteins as we detected selective proteolysis also for internal sites. The structure of the protein dictates the site and extent of cleavage that leads to specific pruning of disordered regions and retention of folded regions. It will be interesting to determine in future studies how the processing of each of our identified processed proteins modulates its function and consequently affect downstream processes.

In summary, our newly introduced method offers an innovative approach for uncovering the substrate landscape of the 20S proteasome. While the PiP-MS technique does not provide direct insight into cellular status due to its application to cell lysates, it enables the investigation of how cellular growth conditions and proteasome variants impact the 20S substratome. This versatility suggests potential applications in various areas, such as profiling of the substrate landscape of the immunoproteasome and intermediate proteasomes. PiP-MS can also illuminate on the influence of proteasome activators like PA28αβ, PA28γ, and PA200 on proteins targeted for 20S proteasome degradation, and on the substrate specificity of regulators like the CCR family that inhibit 20S proteasome function (Olshina et al, 2020). It holds the potential to elucidate the impact of cellular growth conditions such as heat-shock, starvation, or UV damage on the 20S substratome, while also uncovering differences between diverse cell types. Furthermore, it is anticipated to play a role in advancing the development of targeted 20S proteasome inhibitors and activators, thereby presenting promising prospects for future therapeutic interventions.

# Methods

## Cells and culture

HEK293T cells obtained from the lab of Eitan Reuveny (Weizmann Institute of Science), were maintained in Dulbecco's Modified Eagle Medium (4.5 g/L D-glucose), supplemented with 10% fetal calf serum, 1% penicillin–streptomycin, 1% sodium pyruvate, 1% nonessential amino acids (Biological Industries), and MycoZap (Lonza cat. # VZA-2032) (DMEM + 5).

HEK293T cells stably expressing the C-terminally FLAG-tagged PSMB2 subunit, produced by the lab of Chaim Kahana (Weizmann Institute of Science), were maintained in DMEM + 5, supplemented with 1 mg/mL puromycin (Sigma cat. #P8833).

HCT-116 colon cancer cell line from the NCI-60 Human Tumor Cell Lines (National Cancer Institute, USA), were grown in RPMI medium, supplemented with 10% fetal calf serum, 1% penicillin–streptomycin, and 1% glutamine.

## Preparation of FLAG-tagged proteasomes

FLAG-tagged proteasomes were prepared as in (Ben-Nissan et al, 2019). In brief, HEK293T, cells stably expressing the C-terminally FLAG-tagged PSMB2-subunit, were grown in twenty 15-cm plates. After harvesting, lysis was performed in buffer containing 50 mM

Tris pH 7.6, 150 mM NaCl, and 0.5% NP-40, on ice via tissue grinding, and then proteasomes were isolated using M2 FLAG Affinity Gel (A2220; Sigma). Before elution, beads with the proteasomes were incubated for 1.5 h in lysis buffer supplemented with 0.5 M NaCl, in order to remove the 19S proteasome subunits. Subsequently, proteasomes were washed with TBS, and eluted in TBS buffer containing 0.5 mg/ml FLAG peptide, concentrated and flash-frozen.

## Induction of oxidative stress

Cells were grown to 70% confluency in their regular medium and then the medium was replaced to DMEM without sodium pyruvate and MycoZap, and supplemented with 100 μM DEM (Sigma cat. #D2650) (prepared fresh) for 17 h. Cells were then harvested by trypsinization and washed with PBS before flash freezing.

## Denaturing lysates

Cell pellets were lysed in RIPA buffer (20 mM Tris pH 7.6, 150 mM NaCl, 1 mM EDTA, 1% NP-40, 1% Na-deoxicholate, 0.26 mM PMSF, 1 mM Benzamidine and 1.4 μg/ml Pepstatine). Lysates were incubated on ice for 10 min, centrifuged at 4 °C for 10 min, at $16,000 \times g$, and the supernatant was collected. Total protein concentration was estimated by Bradford assay.

## Native lysates

HEK293T, grown to confluency, were harvested by trypsinization and washed with PBS to remove excess trypsin, before pelleting and flash freezing. On the day of the digestion experiment, a single pellet containing around 10 million cells was thawed and 750 μl of buffer, along with 7.5 μl of 0.1 M PMSF, was added. The cells were re-suspended and lysed by five rounds of freeze/thaw and two rounds of gentle bath sonication. The lysate was passed through a 25-gauge needle for homogenization and pelleted at $10,000 \times g$ for 10 min. The supernatant was removed and the total protein concentration was measured by Bradford assay.

## Preparation of samples for PiP-MS

FLAG proteasomes were thawed on ice and their concentration was also re-measured via Bradford assay. The proteasomes and native lysates were mixed for final concentrations of 0.7 mg/mL of proteasomes to 2 mg/mL of lysate. For proteasomes inhibited by epoximicin, they were pre-incubated at 37 °C for 15 min. All reactions were then mixed gently and centrifuged before incubation for 20 h overnight at 37 °C. The following morning, M2 FLAG Affinity Gel was rinsed thoroughly with HEPES buffer and mixed with lysate in a 1:1.25 ratio (V/V) of lysate:beads. Reactions were shaken at 4 °C for 2 h before pouring into a small spin column. Samples were centrifuged to remove the lysate, which was immediately diluted with 10 M urea for a final concentration of 8 M urea. Samples were flash-frozen before later analysis.

## Western blot analysis

Samples were separated on SDS-PAGE, and were further transferred to 0.45 μm immobilon-P PVDF membranes (Millipore),

preactivated in methanol, in Tris-Glycine transfer buffer (pH 8.3) supplemented with 20% methanol for 2.5 h at 400 mA. Membranes were blocked in 5% skim milk powder in TBS-T for 1 h, and incubated with primary antibodies at 4 °C overnight. Membranes were washed three times for 10 min in TBS-T, followed by incubation with appropriate secondary antibodies for 1 h at room temperature. Membranes were again washed three times for 10 min in TBS-T, and developed using WesternBright ECL (Advansta) in myECL Imager (Thermo Scientific) according to the manufacturer's instructions. For western blot analysis, 20 μg of total protein was loaded for each sample.

Primary antibodies used for western blots include anti-FLAG (1:2500, F3165; Sigma), anti-p53 HRP (1:2500, HAF1355; R&D Systems), anti-PSMA1 (1:1000, ab140499; Abcam), anti-α-tubulin (1:10,000, ab184613; Abcam), anti-ubiquitin (1:1000, PW0930; Enzo), anti-NQO2 (1:500, sc-271665; Santa Cruz), anti-NQO1 (1:1000, ab34173; Abcam). Secondary antibodies used for western blots include goat anti-mouse IgG-HRP (1:10,000, 115-035-003; Jackson) and goat anti-rabbit IgG-HRP (1:10,000, 111-035-003; Jackson). Western blots were imaged using Cytiva Amersham ImageQuant 800 V.2.0.0.

## Imaging of MV151-labeled proteasomes

Purified proteasomes were incubated in the presence of 1 μM MV151 (synthetized in-house, in the Medicinal Chemistry Unit, The Nancy and Stephen Grand Israel National Center for Personalized Medicine, Weizmann Institute of Science), for 1.5 h at 37 °C. Following separation on SDS-PAGE, gels were imaged by Cytiva Amersham ImageQuant 800 V.2.0.0, using the manufacturer's setup for Cy3.

## α-synuclein degradation assay

To monitor the degradation capacity of the different 20S proteasomes, 1 pmol of the different proteasomes were incubated with 25 pmol of α-synuclein a final volume of 56 μl, in 50 mM HEPES buffer, pH 7.5, at 37 °C. Ten μl samples were collected every 30 min for 2 h, quenched by 2.5 μl reducing sample buffer and snap-frozen in liquid nitrogen. Samples were then boiled for 5 min, and separated on 15% SDS-PAGE gels. Gels were stained with InstantBlue Coomassie protein stain (ab119211; Abcam), and changes in the level of α-synuclein were quantified by densitometry, and normalized to T0.

## Software

Bar and line graphs were prepared using Microsoft Office Excel 2016 and GraphPad Prism V. 9.5.1 (733). Densitometry analysis was done using ImageJ (v1.47; NIH).

## Proteomics—sample preparation

Proteins were reduced by incubation with 1,4-Dithiothreitol (final concentration of 12 mM) for 30 min at 37 °C and alkylated by incubation with iodoacetamide (final concentration of 40 mM) for 45 min at room temperature in the dark. Samples were diluted with 0.1 M ammonium bicarbonate to a final 2 M urea concentration. Proteins were digested overnight with sequencing-grade porcine

trypsin (Promega) at an enzyme:substrate ratio 1:100 at 37 °C with constant shaking (800 rpm). For GluC digestion samples, GluC was added in an enzyme:substrate ratio 1:100 instead of the trypsin. The digestion was stopped by adding formic acid to a final concentration of 1% (pH <3). The peptide mixtures were loaded onto 96 wells elution plates (Waters), desalted, and eluted with 80% acetonitrile, 0.1% formic acid. After elution, peptides were dried in a vacuum centrifuge, resolubilised in 0.1% formic acid to final 1 mg/ml concentration, and analyzed by MS.

## Proteomics—data acquisition

Samples were analyzed on an Orbitrap Exploris Mass Spectrometer (Thermo Fisher) equipped with a nano-electrospray ion source and a nano-flow LC system (Easy-nLC 1200, Thermo Fisher). Peptides were separated on a 40 cm × 0.75 µm i.d. column (New Objective, PF360- 75-10-N-5) packed in-house with 3-µm C18 beads (Dr. Maisch Reprosil-Pur 120). Buffer A was 0.1% FA (Carl Roth GmbH), and buffer B was 99% ACN (Fisher Scientific A955-212) 0.1% FA (Carl Roth GmbH). Fractionation was achieved with a linear gradient from 5% to 35% buffer B over 120 min, followed by 5 min with an isocratic constant concentration of 90% buffer B. The flow rate was 300 nl/min, and the column was heated to 50 °C. Aliquots of 2 µl of each sample were injected independently and measured in data-independent acquisition mode. The DIA-MS method consisted of a survey MS1 scan from 350 to 2000 *m/z* at a resolution of 120,000 with an AGC target of 50% or 100-ms injection time, followed by DIA in 41 variable-width isolation windows. Precursors were isolated by a quadrupole and fragmented with HCD with a collision energy of 28%. DIA-MS2 spectra were acquired with a scan range of 200 to1800 *m/z* at an orbitrap resolution of 30,000 with an AGC target of 200% or 54-ms injection time.

## Proteomics—data analysis

The data was searched in Spectronaut version 15.10 (Biognosys) (Bruderer et al, 2015) using the direct DIA Pulsar search using the default setting and Trypsin or GluC digestion rule (depending on the experiment) or set to semi-specific rule. The data was searched against the canonical Uniprot fasta database (downloaded March, 2020). The targeted data extraction was performed in Spectronaut version 15.10 with default settings except for the machine learning which was set to "across experiment" and the data filtering which was set to "Qvalue" and data normalisation that was set to median normalisation. Minor grouping was set to Modified peptide sequence. The FDR was set to 1% on peptide and protein level. Peptide and protein-level quantification was exported for further analysis.

## Identification of substrates

To identify significantly degraded proteins, differential analysis was performed comparing peptide quantities (PEP.Quantity), using the moderated *t* test (protti R-package (Quast et al, 2022)), comparing the peptide abundance in the condition with and without the addition of proteasome, followed by Benjamini–Hochberg *P* value correction (Benjamini and Hochberg, 1995). The minimum peptide

length was established at seven amino acids, considering that shorter peptides are often non-proteolytic and would be automatically excluded from the final analysis. This raises the possibility that degradation products of the 20S proteasome could potentially escape detection, either due to their length or the removal of arginine or lysine residues through proteasomal cleavage, hindering their ionization.

Peptides were considered significantly changing if the adjusted *P* value was lower than 0.01 and the absolute value of log2 (FC) was larger than 1. Peptides were further considered to be significantly changing if the peptide completely disappeared from the sample, such that it was detected in all replicates of control condition, while not detected in any condition with proteasome. For protein-level analysis, only proteins with at least three detected peptides were considered. Proteins were identified as a target of proteasome if at least 50% of detected peptides showed significant degradation or, less than 50% of the peptides showed degradation, but the significantly changing peptides were close in sequence and not randomly distributed throughout the sequence. The evaluation of clustering of the significantly changing peptides was performed using evaluate Proteoform Location function in the CCprofiler R-package (Heusel et al, 2019), and considering all the proteins with the calculated *P* value < 0.05. Proteins were considered completely digested if at least 75% of all detected peptides showed significant degradation, otherwise the protein was considered processed. The overlapping substrates across different conditions were analyzed and displayed using Upset R-package (Gomez et al, 2021). In Fig. 7, the processed proteins considered were proteins with less than 75% peptides digested and the *P* value < 0.05 (from CCprofiler R-package).

## Characterization of proteasomal targets

Protein characteristics based on protein sequence (protein isoelectric point (pI), charge, protein length, hydrophobicity, aliphatic index (aIndex), hydrophobic moment and percentages of individual amino acids or amino acid groups) were calculated using Peptides package in R. The following predictions: SCRIBER score (Zhang and Kurgan, 2019), DRNApredDNAscore (Yan and Kurgan, 2017), DRNApredRNAscore (Yan and Kurgan, 2017), MoRFchibi (Disfani et al, 2012), DFLpred (Meng and Kurgan, 2016), DisoRNAscore, DisoPROscore, DisoDNAscore and Accessible surface area (ASA) were downloaded from DescribeProt (Zhao et al, 2021a). Protein localization annotation was downloaded from Uniprot database. Average secondary structure prediction (strand_average, coil_average and helix average) and average disorder values were predicted using DISOPRED3 (Jones and Cozzetto, 2015). To identify whether two groups (degraded vs nondegraded) of proteins significantly differ in certain protein characteristics, *t* test was performed, followed by Benjamini–Hochberg correction (Benjamini and Hochberg, 1995). The difference between mean values for the two groups was calculated, always subtracting the control value from the test group to calculate whether the characteristics was higher or lower in the test group.

## GO-enrichment analysis

We performed functional enrichments of degraded proteins using the *topGO*-package in R (Alexa, 2006). We downloaded current

annotation files for *human*. To focus on the most informative terms, we tested for enrichment with Fisher's exact tests using the weight01-algorithm in *topGO* (Alexa, 2006). We performed the test for Molecular functions. Only terms with adjusted *P* value < 0.01 after the Benjamini–Hochberg *P* value correction (Benjamini and Hochberg, 1995) were considered, however *P* values and not adjusted *P* values are displayed.

## Identification of N- or C-terminal degradation

To identify whether we observe significant degradation of only C- or N-terminal part of the protein, we firstly identified significant abundance changes on peptide level. Differentially abundant peptides were identified using the moderated *t* test (protti R-package), comparing the peptide abundance in the condition with and without the addition of proteasome, followed by Benjamini–Hochberg *P* value correction. Peptides were considered significantly changing if the adjusted *P* value was lower than 0.01 and log2 (FC) was lower than −1, indicating a significant decrease in peptide abundance upon proteasome addition. Only proteins with at least three significantly changing and three nonchanging peptides were considered for further analysis. Peptides were mapped to protein sequences to calculate the peptide position (the position of the amino acid in the middle of the peptide). Two-sided *t* test was performed for each protein to identify whether the positions of degraded peptides were significantly different from the positions of nonchanging peptides. The *P* values were corrected for multiple hypothesis testing using Benjamini–Hochberg *P* value correction. The protein was considered differentially degraded if the degraded peptides had significantly higher or lower mean position (*P* value < 0.05).

## Data availability

Data supporting the findings of this work are available within the paper and its Extended View (Figures and Tables) files. The MS-based proteomics data have been deposited to the ProteomeXchange Consortium via the PRIDE partner repository and are available via ProteomeXchange with the identifier PXD044399.

## Peer review information

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

## Acknowledgements

We would like to thank Dr. Helaneh Salameh, Dr. Khriesto Shurrush, and Dr. Haim Barr from the Medicinal Chemistry Unit in the Nancy & Stephen Grand Israel National Center for Personalized Medicine, Weizmann Institute of Science, for synthesizing the proteasome activity probe MV151. M Sharon is grateful for the support of an Advanced Grant European Research Council (ERC) (Horizon 2020)/ERC Grant Agreement no. 101092725. M Sharon is the incumbent of the Aharon and Ephraim Katzir Memorial Professorial Chair.

## Author contributions

**Monika Pepelnjak**: Data curation; Formal analysis; Writing—original draft.
**Rivkah Rogawski**: Data curation; Formal analysis. **Galina Arkind**: Data curation.
**Yegor Leushkin**: Data curation. **Irit Fainer**: Data curation. **Gili Ben-Nissan**: Data

curation; Formal analysis; Writing—review and editing. **Paola Picotti**: Conceptualization; Supervision; Funding acquisition; Writing—review and editing. **Michal Sharon**: Conceptualization; Supervision; Funding acquisition; Writing—original draft; Writing—review and editing.

## Disclosure and competing interests statement

The authors declare no competing interests.

# Expanded View Figures

**Figure EV1.  Analysis of significantly changing peptides in PiP-MS.**

(**A**) Volcano plot shows significantly changing peptides of proteasomal cleavage, followed by GluC degradation. Each point represents a peptide measured. The color indicates whether the change is indicative of reduction in abundance due to degradation (red), increase in abundance (dark gray) or no significant change (light gray). (**B**) String network (Szklarczyk et al, 2021) of proteins that showed increased abundance upon addition of proteasome (three or more significantly changing peptides). Most peptides originate from the proteasome complex. The lines show confident physical interactions (confidence cutoff >0.7). (**C, D**) The plot shows degradation of specific proteins selected for validation. Each line represents a peptide at a specific amino acid position. The color indicates whether the peptide showed significant signs of degradation (red and blue) or not (gray). Namely, peptides decreasing in abundance (dark red), specific peptides completely disappearing upon addition of proteasome (blue) or new semi-specific (proteasome-specific) peptides appearing (light red).

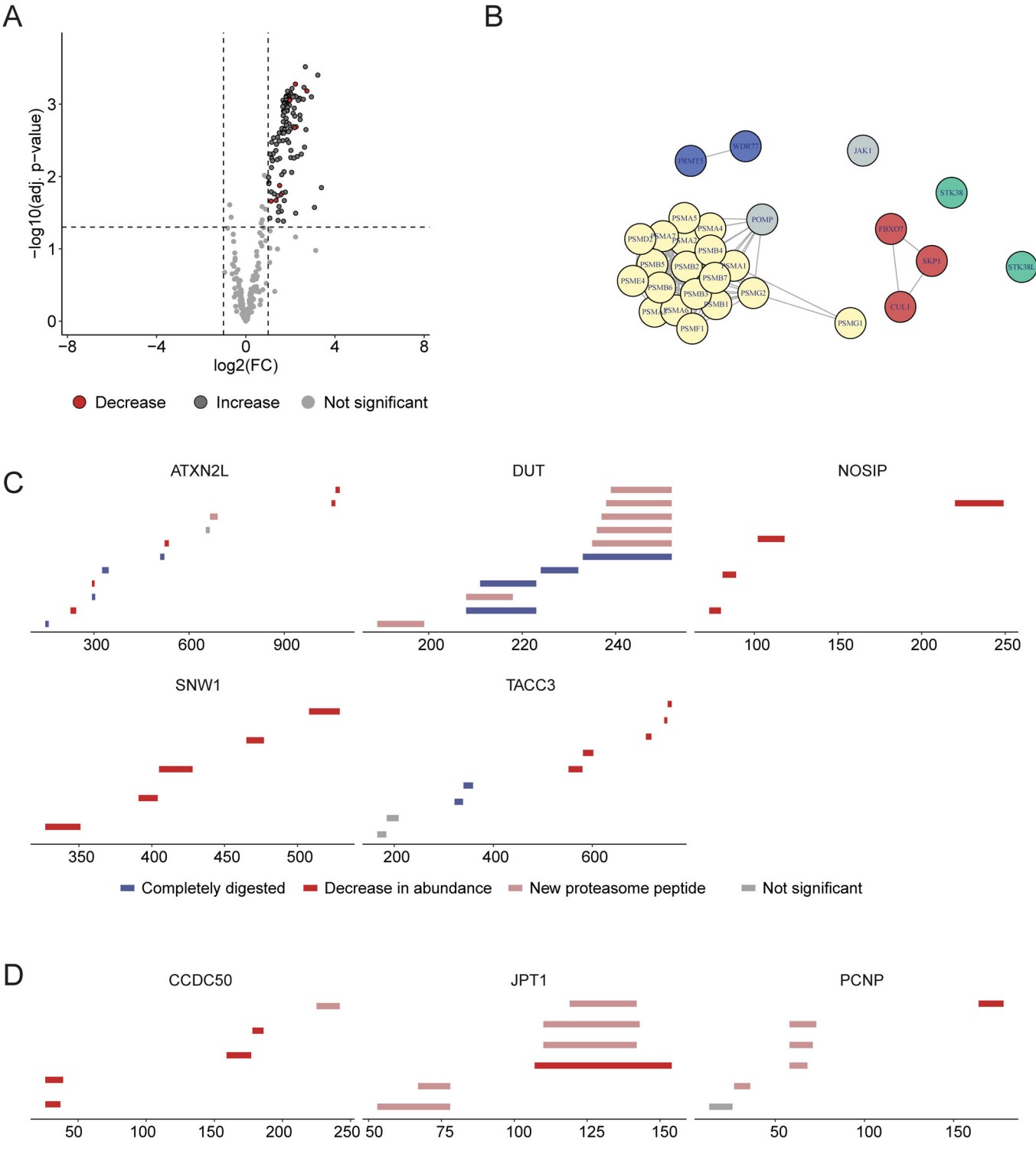

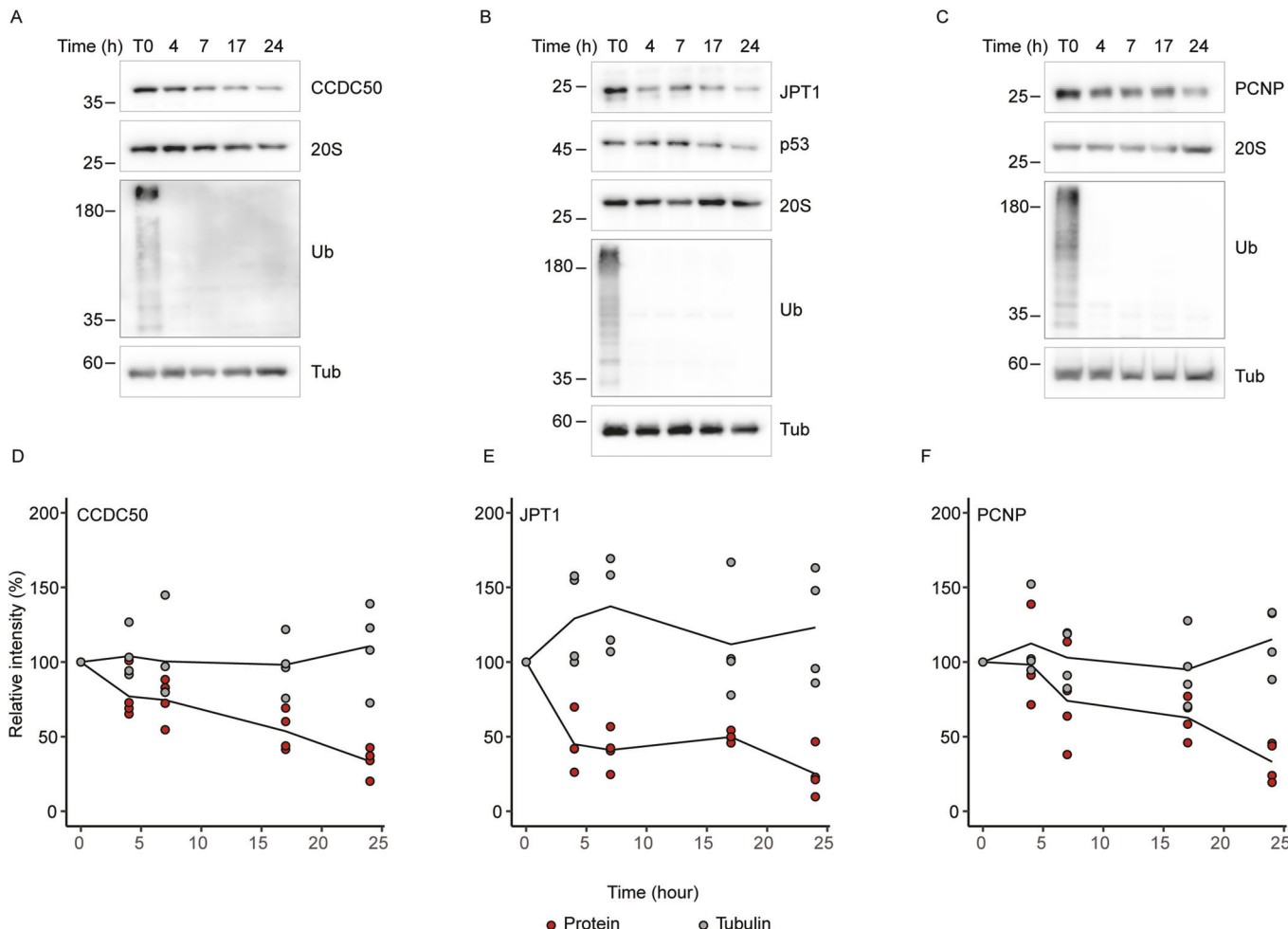

**Figure EV2.  Cycloheximide chase assay for monitoring the cellular stability of the PiP-MS hits.**

Representative western blots showing the 20S proteasome-dependent degradation of (**A**) CCDC50, (**B**) JPT1 and (**C**) PCNP. Cells were treated with the ubiquitination inhibitor, TAK-243 together with cycloheximide, to inhibit protein synthesis. Cells were harvested after 4, 7, 17 and 24 h. Stabilities of the target proteins (FLAG-tagged) and the 20S proteasome were analyzed by western blots using antibodies against FLAG, p53 and PSMA1 (a 20S proteasome subunit). To monitor the inactivation of the ubiquitination cascade an anti-ubiquitin antibody was used. p53 and Tubulin (Tub) were used a positive and negative controls, respectively. Changes in the levels of the (**D**) CCDC50, (**E**) JPT1 and (**F**) PCNP relative to the initial time point were quantified from four independent experiments. Each point corresponds to an individual experiment, with red dots indicating the analyzed protein and gray dots representing tubulin, the control. Source data are available online for this figure.

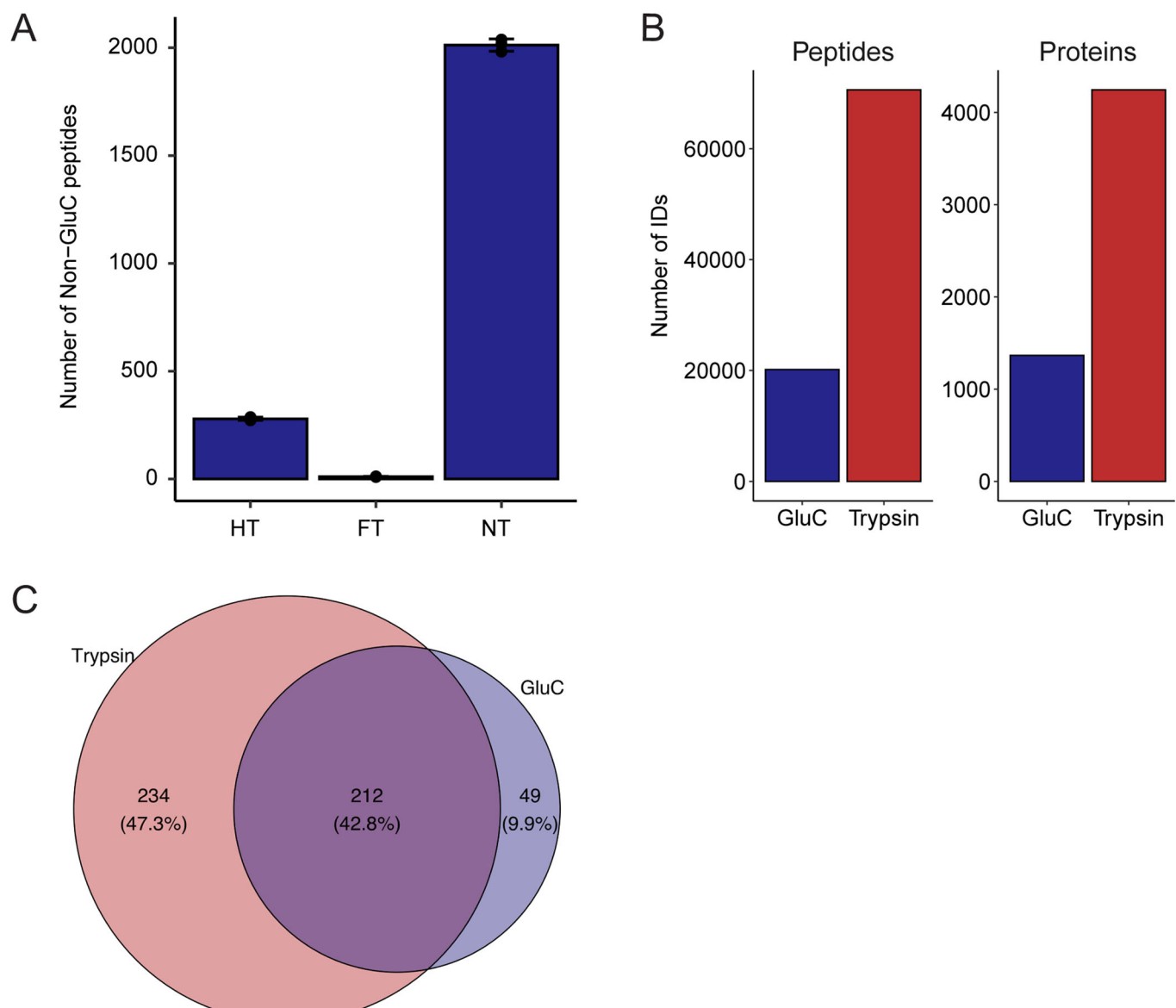

**Figure EV3.   Assessing the efficiency of the second digestion step using GluC or trypsin.**

(A) Analysis of proteasome unique peptides in the GluC experiment reveal that most peptides have no tryptic ends (NT). Fully-tryptic (FT) peptides have two tryptic ends, and half-tryptic (HT) peptides have one tryptic end. Error bars represent the mean $+/-$ SD of $n = 3$ replicates. (B) Number of detected peptides and proteins when Trypsin or GluC was used in the second processing step of PiP-MS experiment. (C) Number of degraded proteins identified with two different enzymes. Only proteins detected in both datasets were considered in this analysis.

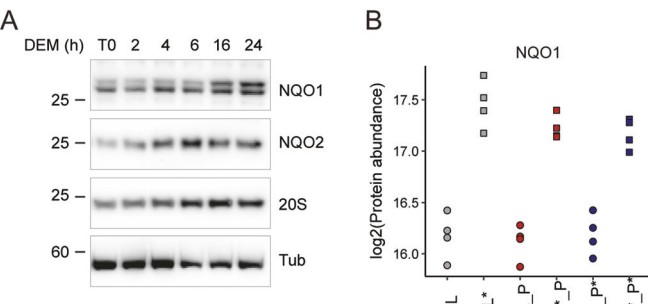

**Figure EV4.   Induction of oxidative stress by treating cells with DEM.**

(A) Following the exposure of cells to DEM, western blot analysis was used for monitoring the levels of NQO1, NQO2 and an 20S proteasome subunit (20S, PSMA1). Tubulin (Tub) served as a control. As expected, induction of oxidative stress led to an increase in NQO1, NQO2 and 20S proteasome levels. The experiment was repeated three times. (B) LC/MS proteomic analysis similarly indicates an upregulation of NQO1 under conditions of oxidative stress (labeled with an asterisk for Lysate (L) and Proteasome (P)) relative to naïve conditions. Each point corresponds to an individual experiment. Source data are available online for this figure.

A

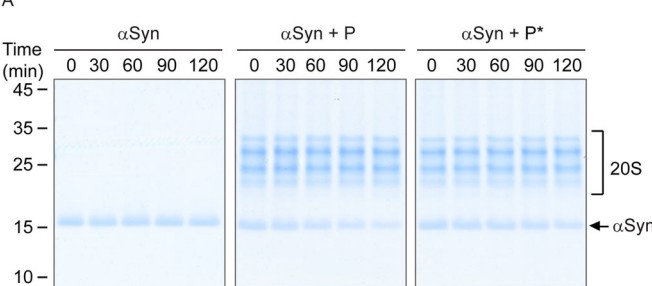

B

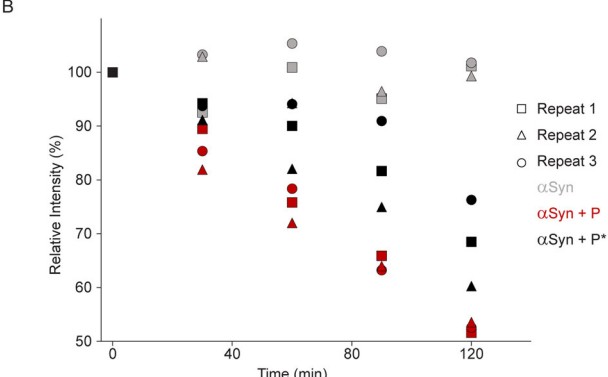

**Figure EV5. The naïve proteasome more efficiently degrades α-synuclein compared to its oxidized counterpart.**

(A) Representative time-dependent degradation assays of α-synuclein (αSyn) in the presence of naïve (P) and oxidized (P*) 20S proteasomes. As a control, the stability of α-synuclein was measured in the absence of the proteasome. (B) Raw data for Fig. 4C. Squares, triangles and circles represent biological repeats. Gray, red and black data points denote the levels of α-synuclein (αSyn) over time in the absence or presence of naïve (P) and oxidized (P*) 20S proteasomes. Source data are available online for this figure.

