## [Peer Review File · Molecular Systems Biology]

Systematic Identification of 20S Proteasome Substrates

Monika Pepelnjak, Rivkah Rogawski, Galina Arkind, Yegor Leushkin, Irit Fainer, Gili Ben-Nissan, Paola Picotti, and Michal Sharon

DOI: 10.15252/msb.202312000

Corresponding author(s): Michal Sharon (Michal.Sharon@weizmann.ac.il) , Paola Picotti (picotti@imsb.biol.ethz.ch)

Review Timeline:	Submission Date:	13th Sep 23
	Editorial Decision:	19th Oct 23
	Revision Received:	13th Dec 23
	Accepted:	5th Jan 24

Editor: Maria Polychronidou

Transaction Report:

19th Oct 2023

Manuscript Number: MSB-2023-12000

Title: Systematic Identification of 20S Proteasome Substrates

Dear Michal,

Thank you again for submitting your work to Molecular Systems Biology. Unfortunately, despite several reminders, we have not received a report from reviewer #2. In the interest of time and given that the recommendations of reviewers #1 and #3 are similar, we decided to proceed with making a decision based on the two available reports. As you will see below, the two reviewers are positive about the study. However, they raise a series of concerns, which we would ask you to address in a revision.

The comments of the referees are rather clear and seem straightforward to address so I think there is no need to repeat any of them here. All issues raised by the referees would need to be satisfactorily addressed. Please let me know in case you would like to discuss in further detail any of the issues raised, I would be happy to schedule a call. Of course, it is not problem to extend the revision deadline.

On a more editorial level, we would ask you to address the following points:

- Please provide a .doc version of the manuscript text (including legends for the main figures) and individual production quality figure files for the main Figures (one file per figure).
- Please include 5 keywords.
- The abstract is rather long, please shorten it according to our author guidelines (i.e. ~175 words).
- We have replaced Supplementary Information by the Expanded View (EV format). In this case, all additional figures can be provided as EV Figures. Please provide one file per EV Figure. Their legends should be included in the manuscript text. For detailed instructions regarding expanded view please refer to our Author Guidelines: .
- Tables S1-S4 should be provided and called out in the text as Datasets EV1-EV4. Please provide one file per EV Dataset. Each Please include the description of each EV Dataset in the dataset file itself, i.e. in a separate tab for .xls files or as a README.txt file in .zip folders for .csv files.
- Please provide a "standfirst text" summarizing the study in one or two sentences (approximately 250 characters), three to four "bullet points" highlighting the main findings and a "synopsis image" (550px width and max 400px height, jpeg format) to highlight the paper on our homepage.
- Please include a "Disclosure and Competing Interests statement" in the main text.
- All Materials and Methods need to be described in the main text. We would encourage you to use 'Structured Methods', our new Materials and Methods format. According to this format, the Material and Methods section should include a Reagents and Tools Table (listing key reagents, experimental models, software and relevant equipment and including their sources and relevant identifiers) followed by a Methods and Protocols section in which we encourage the authors to describe their methods using a step-by-step protocol format with bullet points, to facilitate the adoption of the methodologies across labs. More information on how to adhere to this format as well as downloadable templates (.doc or .xls) for the Reagents and Tools Table can be found in our author guidelines: . An example of a Method paper with Structured Methods can be found here:
- Please include a Data availability section describing how the data, code etc. have been made available. This section needs to be formatted according to the example below:
The datasets and computer code produced in this study are available in the following databases:
 - Chip-Seq data: Gene Expression Omnibus GSE46748 (<https://www.ncbi.nlm.nih.gov/geo/query/acc.cgi?acc=GSE46748>)
 - Modeling computer scripts: GitHub (<https://github.com/SysBioChalmers/GECKO/releases/tag/v1.0>)
 - [data type]: [full name of the resource] [accession number/identifier] ([doi or URL or identifiers.org/DATABASE:ACCESSION])
- For data quantification: please specify the name of the statistical test used to generate error bars and P values, the number (n) of independent experiments (specify technical or biological replicates) underlying each data point and the test used to calculate

p-values in each figure legend. The figure legends should contain a basic description of n, P and the test applied. Graphs must include a description of the bars and the error bars (s.d., s.e.m.).

- The References should be formatted according to the Molecular Systems Biology reference style (i.e. ordered alphabetically and listing the first 10 authors followed by et al).

- When you resubmit your manuscript, please download our CHECKLIST (<https://bit.ly/EMBOPressAuthorChecklist>) and include the completed form in your submission.

Please note that the Author Checklist will be published alongside the paper as part of the transparent process (<https://www.embopress.org/page/journal/17444292/authorguide#transparentprocess>).

If you feel you can satisfactorily deal with these points and those listed by the referees, you may wish to submit a revised version of your manuscript. Please attach a covering letter giving details of the way in which you have handled each of the points raised by the referees. A revised manuscript will be once again subject to review and you probably understand that we can give you no guarantee at this stage that the eventual outcome will be favorable.

Kind regards,

Maria

Maria Polychronidou, PhD
Senior Editor
Molecular Systems Biology

We realize that it is difficult to revise to a specific deadline. In the interest of protecting the conceptual advance provided by the work, we recommend a revision within 3 months (17th Jan 2024). Please discuss the revision progress ahead of this time with the editor if you require more time to complete the revisions. Use the link below to submit your revision:

IMPORTANT: When you send your revision, we will require the following items:

1. the manuscript text in LaTeX, RTF or MS Word format
2. a letter with a detailed description of the changes made in response to the referees. Please specify clearly the exact places in the text (pages and paragraphs) where each change has been made in response to each specific comment given
3. three to four 'bullet points' highlighting the main findings of your study
4. a short 'blurb' text summarizing in two sentences the study (max. 250 characters)
5. a 'thumbnail image' (550px width and max 400px height, Illustrator, PowerPoint or jpeg format), which can be used as 'visual title' for the synopsis section of your paper.
6. Please include an author contributions statement after the Acknowledgements section (see <https://www.embopress.org/page/journal/17444292/authorguide>)
7. Please complete the CHECKLIST available at (<https://bit.ly/EMBOPressAuthorChecklist>). Please note that the Author Checklist will be published alongside the paper as part of the transparent process (<https://www.embopress.org/page/journal/17444292/authorguide#transparentprocess>).
8. When assembling figures, please refer to our figure preparation guideline in order to ensure proper formatting and readability in print as well as on screen: <https://bit.ly/EMBOPressFigurePreparationGuideline>
See also figure legend guidelines: <https://www.embopress.org/page/journal/17444292/authorguide#figureformat>
9. Please note that corresponding authors are required to supply an ORCID ID for their name upon submission of a revised manuscript (EMBO Press signed a joint statement to encourage ORCID adoption). (<https://www.embopress.org/page/journal/17444292/authorguide#editorialprocess>)
Currently, our records indicate that the ORCID for your account is 0000-0003-3933-0595.

Link Not Available

The system will prompt you to fill in your funding and payment information. This will allow Wiley to send you a quote for the article processing charge (APC) in case of acceptance. This quote takes into account any reduction or fee waivers that you may be eligible for. Authors do not need to pay any fees before their manuscript is accepted and transferred to the publisher.

EMBO Press participates in many Publish and Read agreements that allow authors to publish Open Access with reduced/no publication charges. Check your eligibility: <https://authorservices.wiley.com/author-resources/Journal-Authors/open-access/affiliation-policies-payments/index.html>

*** PLEASE NOTE *** As part of the EMBO Press transparent editorial process initiative (see our Editorial at <https://dx.doi.org/10.1038/msb.2010.72>), Molecular Systems Biology publishes online a Review Process File with each accepted manuscripts. This file will be published in conjunction with your paper and will include the anonymous referee reports, your point-by-point response and all pertinent correspondence relating to the manuscript. If you do NOT want this File to be published, please inform the editorial office at msb@embo.org within 14 days upon receipt of the present letter.

Reviewer #1:

Pepelnjak et al have taken a very thorough and novel approach to identifying the 20S proteasome substrates with HEK293 cells. Adapting an approach of limited proteolysis (normally using proteinase K) with 20S proteasome was a clever to address a critical question in the proteasome field; what are the substrates of the 20S proteasome? The authors do a lot of great analysis that I believe will be very important for the field at large. This is a high quality paper and I have only clarifying questions/concerns.

- 1) I believe the authors have made a good attempt at describing this approach. It would be very important to improve some of the descriptions of what was actually done.
 - a. How were the MS spectra analyzed to generate the resulting final data set that yields a "peptide sequence with the associated change"? Also, please describe what you mean by "associated change".
 - b. In the model, the authors show preservation of a peptide or I guess a new peptide with proteasome treatment. This model is not that clear. It would be nice to see the whole process through for at least one or two example proteins. Draw it out in detail.
 - c. If a peptide is different in the proteasome treated sample then that protein was deemed a 20S substrate, correct? Again, a beginning to end depiction of the analysis with details would be great.
 - d. Is the "peptide with associated change" the location of where the proteasome would cleave or the peptide that is lost with proteasome?
 - e. Why does proteasome primarily cause a decrease in peptides? It seems that with a decrease in one peptide there should be a corresponding increase in a new peptide. I calculated that there are around 150 proteins that show significant decrease or complete loss of associated peptides along with an increase in peptides. Are these not some of the more likely proteasome substrates? These proteins may be the better substrates as they are yielding new peptides that are cleaved by the proteasome. It also may reveal the protein substrates that yield relevant peptides for signaling as suggested by ref. 37.
 - f. Again, Is the calculated FC based on the control, then is the peptide going down just the one presented. Why would only one peptide of a whole protein being degraded by the proteasome go down. I would expect the best substrates to be the one that have a lot of new peptides that go up and some peptides that go down and some that do not change at all.
 - g. The authors should provide the complete Excel data sheets with all the analysis on the sheet so readers can see how the protein substrates were identified using this method.
 - h. The E1 inhibitor did not seem to lead to an increase in ubiquitination? This seems a bit odd. Please provide some explanation.
 - i. Quantify the aSyn gels.

Reviewer #3:

Proteasomal degradation is vital in all cells and organisms, and dysfunction or failure of proteasomal degradation is associated with various human diseases, including cancer and neurodegeneration. What is the contribution of the free 20S proteasome over that of the 26S proteasome remains elusive. This is mainly due to the difficulty of separating the function of the 20S proteasome from that of the 26S proteasome. Here, the authors have adapted the limited proteolysis-mass spectrometry (LiP-MS) method to identify 20S proteasome substrates. It is based on the detection of putative 20S proteasome cleavage sites by MS, a method referred to as proteasomal induced proteolysis (PIP)-MS. Using this method, they have identified hundreds of putative 20S proteasome substrates, which are enriched in RNA/DNA-binding proteins that contains intrinsically disordered regions (IDRs), as previously reported in the literature. The novelty comes from the discovery that loss of conformation upon oxidative stress is not the main triggers for 20S-mediated degradation, as the selectivity of 20S proteasomes remains poorly affected by oxidative stress. They further showed that the function of oxidised 20S proteasomes is hampered compared to native 20S proteasome which may explain why cells adapt to oxidative stress by making more proteasomes in a later stage. The catalytic activity of oxidised 20S proteasomes was unaffected by oxidative stress, advocating for an alternative mechanism of proteasome inhibition yet to be discovered. Finally, they found that some 20S proteasome substrates are only partially degraded

forming truncated proteins, which may have an altered function.

While it is already known that substrates of the 20S proteasomes are mainly IDR-containing proteins, the work by Pepelnjak and Rogawski et al., does bring significant new advances to the proteasome field:

- (1) So far there is no methodology to interrogate what are the substrates of free 20S proteasomes in vivo. While the methodology presented by the authors is still in vitro, I think this may be one of the first step toward better understanding substrate specificity of the 20S proteasome. Moreover, PiP-MS can be applied to non-conventional forms of the proteasome such as immunoproteasomes and PA28-capped proteasomes, broadening its use for the scientific community.
- (2) Partial protein degradation by the proteasome may be more frequent than previously expected, even if this has not been confirmed by the authors in a more physiological condition.
- (3) Oxidised 20S proteasomes have reduced degradative capacity while their catalytic sites remained unaffected, the mechanism yet to be discovered.

Data presented in the manuscript support overall authors' conclusion, although some specific conclusions may need to be strengthened (see below). On the whole, the paper is well written, timely and will bring significant technical advances to the proteasome field. In summary, I do think that this paper will be a good addition to the scientific literature, and I would support publication of this manuscript providing the more specific concerns listed below are addressed.

Major points:

- (1) The data presented in Supplementary Fig. 2 are not supporting the authors' conclusion to me: "We observed that these three proteins were degraded in a ubiquitin independent manner in the presence of TAK-243 (Fig. 2E and Supplementary Fig. 2)". No visible decrease of 20S substrates is detected in Supplementary Fig. 2, and it seems even to potentially be higher for CCDC50 and JPT1 at 24h, while the housekeeping gene tubulin remains overall pretty stable. This needs to be clarified.
- (2) The authors should add all individual data points on their histograms/curves.
- (3) Do the authors have data supporting the conclusion that substrates in Figure 2E are degraded by the 20S proteasome and not just cleaved by other proteases, such as co-treatment with epoxomicin?
- (4) This paragraph lacks a bit of clarity to me: "Overall, for 20% of proteins in the human proteome we detected at least three peptide indicative of 20S proteasome cleavage (Fig. 2C). Altered peptides from this set of proteins were either peptides that decreased in intensity (adj. p-value < 0.01, log₂(FC) < -1) , completely disappeared or new semi-specific peptides that appeared upon proteasome treatment (Fig. 2C). Overall we identified 280 candidate substrates of the 20S proteasome from 2180 peptides pinpointing proteasome cleaved regions.". It's not clear how the authors have only detected 280 candidates while they have detected at least three peptide indicative of 20S proteasome cleavage for 20% of proteins in the human proteome (~20,000 different proteins in human cells, so ~4000 proteins with peptide indicative of 20S proteasome cleavage?). Do they mean 20% of MS detected proteins or, as stated, 20% of proteins in the human proteome? This needs to be clarified by clearly stating how many peptides they have identified in total and the corresponding number of proteins. From these numbers how many had at least three peptide indicative of 20S proteasome cleavage, and how this going down to 280 candidates at the end.

Minor points:

- (1) "Supplementary Fig.1C" at the end of page 6 should be "Supplementary Fig.1D".
- (2) Figure 6A: if "Ctrl/Ox" is referring to lysate, the authors should keep the previous nomenclature "L/L*". What Ctrl, Ox, P and P* mean should also be mentioned in the legend.
- (3) Is partial degradation by the 20S proteasome occurring due to the presence of a folded domain in the non-degraded part? This could be analysed or at least discussed.
- (4) There is a typo end of page 17: "phenomenon, with more than 200 proteins partially processed? by the complex."

Reviewer 1

Pepelnjak et al have taken a very thorough and novel approach to identifying the 20S proteasome substrates with HEK293 cells. Adapting an approach of limited proteolysis (normally using proteinase K) with 20S proteasome was a clever to address a critical question in the proteasome field; what are the substrates of the 20S proteasome? The authors do a lot of great analysis that I believe will be very important for the field at large. This is a high quality paper and I have only clarifying questions/concerns.

1) I believe the authors have made a good attempt at describing this approach. It would be very important to improve some of the descriptions of what was actually done.

We thank the reviewer for this positive comment, indicating that this is a high-quality paper that will be very important for the field at large. We have carefully studied your comments and suggestions, and addressed all of them, as described below.

a. How were the MS spectra analyzed to generate the resulting final data set that yields a "peptide sequence with the associated change"? Also, please describe what you mean by "associated change".

We acknowledge the reviewer's comment and recognize the need for additional information concerning the MS spectra analysis. Consequently, we have incorporated a detailed explanation of the PiP-MS analysis, along with the inclusion of two schemes illustrating the process (new Figs. 1B-C). To identify 20S proteasome substrates by PiP-MS, peptide abundances for all identified peptides were measured in data-independent mode. A comparison was made between the control and proteasome-treated conditions for all peptides of a given protein. The peptide abundance could either remain unchanged or undergo a significant change between conditions. A peptide was considered significantly changing if a peptide with two protease specific ends (specific peptide) decreased in abundance, if a new peptide with a proteasome cleavage site emerged (semi-specific peptide), or if a specific peptide disappeared entirely upon the addition of the proteasome. To translate peptide-level information into protein-level information, we adhered to a stringent criterion. A protein was considered a substrate if at least 50% of detected peptides showed significant change or when less than 50% showed significant change (processed proteins) but the significantly changing peptides colocalized together in the peptide sequence.

b. In the model, the authors show preservation of a peptide or I guess a new peptide with proteasome treatment. This model is not that clear. It would be nice to see the whole process through for at least one or two example proteins. Draw it out in detail.

As suggested by the reviewer, we have incorporated two schematics (new Figs. 1B-C) along with a textual explanation to elucidate the PiP-MS methodology.

c. If a peptide is different in the proteasome treated sample then that protein was deemed a 20S substrate, correct? Again, a beginning to end depiction of the analysis with details would be great.

The reviewer is correct there are three situations in which a protein was categorized as a substrate:

- The intensity of the peptide exhibited a significant decrease.
- The peptide completely disappeared.
- A new peptide emerged.

To elucidate this aspect, we have added a textual explanation accompanied by a schematic description (new Figs. 1B-C).

d. Is the "peptide with associated change" the location of where the proteasome would cleave or the peptide that is lost with proteasome?

We agree with the reviewer's observation that in Figure EV1, we did not explicitly indicate whether the observed changes in peptides, following the addition of the 20S proteasome, are attributed to a reduction in intensity, complete disappearance, or emergence of new peptides. To address this, we have made modifications to the figure (now presented as new Fig. EV1C-D) and updated the figure legend accordingly.

e. Why does proteasome primarily cause a decrease in peptides? It seems that with a decrease in one peptide there should be a corresponding increase in a new peptide. I calculated that there are around 150 proteins that show significant decrease or complete loss of associated peptides along with an increase in peptides. Are these not some of the more likely proteasome substrates? These proteins may be the better substrates as they are yielding new peptides that are cleaved by the proteasome. It also may reveal the protein substrates that yield relevant peptides for signaling as suggested by ref. 37.

After the cleavage of a protein region, proteasome-specific peptides are produced. However, these peptides may undergo additional degradation by the proteasome or other cellular peptidases, potentially hindering the detection and analysis of resulting degradation products by MS. Moreover, during degradation, peptides may lose lysine or arginine residues, impeding their ionization and subsequent detection by MS. In our analysis, the minimum peptide length was set to 7 amino acids; therefore, if a peptide has a shorter sequence, it would not be recognized by the search engine. The choice of a 7AA length is based on the fact that shorter peptides are often non-proteolytic and would be automatically excluded from the final analysis. We have now revised the text in the method section accordingly.

f. Again, Is the calculated FC based on the control, then is the peptide going down just the one presented. Why would only one peptide of a whole protein being degraded by the proteasome go down. I would expect the best substrates to be the one that have a lot of new peptides that go up and some peptides that go down and some that do not change at all.

Our analysis follows stringent criteria for identifying 20S proteasome substrates. Proteins are classified as substrates only if they have a minimum of 50% of peptides significantly changing or if proteins have significantly changing peptides mapping to the same protein region (for processed proteins with less than 50% changing peptides). We believe that such harsh filtering criteria

allowed us to focus on most interesting targets. In the revised version of the manuscript, we provide a clear description of these filtering criteria.

g. The authors should provide the complete Excel data sheets with all the analysis on the sheet so readers can see how the protein substrates were identified using this method.

We agree with the reviewer and have uploaded the complete Excel data sheets containing all relevant information, allowing readers to thoroughly examine the identification of protein substrates using our method.

h. The E1 inhibitor did not seem to lead to an increase in ubiquitination? This seems a bit odd. Please provide some explanation.

We greatly appreciate the reviewer's attention to detail in spotting an error in our work, for which we sincerely apologize. In Supplementary Figure 2, we accidentally presented control gel images that did not include the addition of the E1 ubiquitination inhibitor, TAK-243 to the cellular samples, instead of the intended representation of the treated samples. We have made the necessary corrections and updated the figure, now referred to as EV2, to accurately reflect the outcomes of the TAK-243 treatment. This revised figure clearly illustrates the inhibitory effects on ubiquitination and the 20S proteasome degradation of CCDC50, JPT1, and PCNP.

i. Quantify the aSyn gels.

We thank the reviewer for the suggestion and have made the necessary modification to Figure EV5 (formerly Supplementary Figure 5) by incorporating the quantification data points for the three biological repeats of the α -synuclein gels.

Reviewer #3

Proteasomal degradation is vital in all cells and organisms, and dysfunction or failure of proteasomal degradation is associated with various human diseases, including cancer and neurodegeneration. What is the contribution of the free 20S proteasome over that of the 26S proteasome remains elusive. This is mainly due to the difficulty of separating the function of the 20S proteasome from that of the 26S proteasome. Here, the authors have adapted the limited proteolysis-mass spectrometry (LiP-MS) method to identify 20S proteasome substrates. It is based on the detection of putative 20S proteasome cleavage sites by MS, a method referred to as proteasomal induced proteolysis (PIP)-MS. Using this method, they have identified hundreds of putative 20S proteasome substrates, which are enriched in RNA/DNA-binding proteins that contains intrinsically disordered regions (IDRs), as previously reported in the literature. The novelty comes from the discovery that loss of conformation upon oxidative stress is not the main triggers for 20S-mediated degradation, as the selectivity of 20S proteasomes remains poorly affected by oxidative stress. They further showed that the function of oxidised 20S proteasomes is hampered compared to native 20S proteasome which may explain why cells adapt to oxidative stress by making more

proteasomes in a later stage. The catalytic activity of oxidised 20S proteasomes was unaffected by oxidative stress, advocating for an alternative mechanism of proteasome inhibition yet to be discovered. Finally, they found that some 20S proteasome substrates are only partially degraded forming truncated proteins, which may have an altered function.

While it is already known that substrates of the 20S proteasomes are mainly IDR-containing proteins, the work by Pepelnjak and Rogawski et al., does bring significant new advances to the proteasome field:

(1) So far there is no methodology to interrogate what are the substrates of free 20S proteasomes in vivo. While the methodology presented by the authors is still in vitro, I think this may be one of the first step toward better understanding substrate specificity of the 20S proteasome. Moreover, PiP-MS can be applied to non-conventional forms of the proteasome such as immunoproteasomes and PA28-capped proteasomes, broadening its use for the scientific community.

(2) Partial protein degradation by the proteasome may be more frequent than previously expected, even if this has not been confirmed by the authors in a more physiological condition.

(3) Oxidised 20S proteasomes have reduced degradative capacity while their catalytic sites remained unaffected, the mechanism yet to be discovered.

Data presented in the manuscript support overall authors' conclusion, although some specific conclusions may need to be strengthened (see below). On the whole, the paper is well written, timely and will bring significant technical advances to the proteasome field. In summary, I do think that this paper will be a good addition to the scientific literature, and I would support publication of this manuscript providing the more specific concerns listed below are addressed.

We thank the reviewer for indicating that the study brings significant new advances to the proteasome field. We have carefully studied the reviewers' comments and suggestions, and addressed all of them, as described below.

Major points:

(1) The data presented in Supplementary Fig. 2 are not supporting the authors' conclusion to me: "We observed that these three proteins were degraded in a ubiquitin independent manner in the presence of TAK-243 (Fig. 2E and Supplementary Fig. 2)". No visible decrease of 20S substrates is detected in Supplementary Fig. 2, and it seems even to potentially be higher for CCDC50 and JPT1 at 24h, while the housekeeping gene tubulin remains overall pretty stable. This needs to be clarified.

We greatly appreciate the reviewer's attention to detail in spotting an error in our work, for which we sincerely apologize. In Supplementary Figure 2, we accidentally presented control gel images

that did not include the addition of the E1 ubiquitination inhibitor, TAK-243 to the cellular samples, instead of the intended representation of the treated samples. We have made the necessary corrections and updated the figure, now referred to as EV2, to accurately reflect the outcomes of the TAK-243 treatment. This revised figure clearly illustrates the inhibitory effects on ubiquitination and the 20S proteasome degradation of CCDC50, JPT1, and PCNP.

(2) The authors should add all individual data points on their histograms/curves.

We thank the reviewer for this comment and have added where possible the individual data points to the histograms and curves. Please see Figures 2D, 2F, 6B, 7C, EV2D-F, EV3A, EV4B and EV5B.

(3) Do the authors have data supporting the conclusion that substrates in Figure 2E are degraded by the 20S proteasome and not just cleaved by other proteases, such as co-treatment with epoxomicin?

We have previously attempted including epoxomicin in the cycloheximide chase cellular experiments involving the ubiquitination inhibitor TAK-243, however the combination of these three inhibitors causes rapid cell death. Therefore, to validate proteasome mediated degradation and rule out cleavage by other proteases, we performed time dependent degradation assays in the presence of the 20S proteasome with and without epoxomicin, as shown in Figure 2D. The results indeed indicate that proteins identified as PiP-MS hits are substrates of the 20S proteasome.

(4) This paragraph lacks a bit of clarity to me: "Overall, for 20% of proteins in the human proteome we detected at least three peptide indicative of 20S proteasome cleavage (Fig. 2C). Altered peptides from this set of proteins were either peptides that decreased in intensity (adj. p-value < 0.01, log₂(FC) < -1), completely disappeared or new semi-specific peptides that appeared upon proteasome treatment (Fig. 2C). Overall we identified 280 candidate substrates of the 20S proteasome from 2180 peptides pinpointing proteasome cleaved regions." It's not clear how the authors have only detected 280 candidates while they have detected at least three peptide indicative of 20S proteasome cleavage for 20% of proteins in the human proteome (~20,000 different proteins in human cells, so ~4000 proteins with peptide indicative of 20S proteasome cleavage?). Do they mean 20% of MS detected proteins or, as stated, 20% of proteins in the human proteome? This needs to be clarified by clearly stating how many peptides they have identified in total and the corresponding number of proteins. From these numbers how many had at least three peptide indicative of 20S proteasome cleavage, and how this going down to 280 candidates at the end.

We acknowledge the reviewer's point regarding the clarity of our phrasing. As a result, we have modified the text to clarify that among the total identified proteins, 20% are categorized as substrates of the 20S proteasome, requiring a minimum of three significantly modified peptides.

Minor points:

(1) "Supplementary Fig.1C" at the end of page 6 should be "Supplementary Fig.1D".

We apologize for this oversight and have corrected the text accordingly.

(2) Figure 6A: if "Ctrl/Ox" is referring to lysate, the authors should keep the previous nomenclature "L/L*". What Ctrl, Ox, P and P* mean should also be mentioned in the legend.

We apologize for this mistake and have corrected the Figure accordingly.

(3) Is partial degradation by the 20S proteasome occurring due to the presence of a folded domain in the non-degraded part? This could be analysed or at least discussed.

We thank the reviewer for raising this important point and have added the relevant information to the text indicating that the folded-region acts as a stop signal for 20S proteasome cleavage.

(4) There is a typo end of page 17: "phenomenon, with more than 200 proteins partially processed? by the complex."

We thank the reviewer for noticing this typo and have corrected it.

5th Jan 2024

Manuscript number: MSB-2023-12000R

Title: Systematic Identification of 20S Proteasome Substrates

Dear Michal,

Thank you again for sending us your revised manuscript. We think that the performed revisions have satisfactorily addressed the issues raised by the reviewers. As such, I am pleased to inform you that your paper has been accepted for publication.

Kind regards,

Maria

Maria Polychronidou, PhD
Senior Editor
Molecular Systems Biology
